# The retreival of snow properties from Sentinel-3 SLSTR - part 2: results and validation

**Linlu Mei[1], Vladimir Rozanov[1], Evelyn Jäkel[2], Xiao Cheng[3], Marco Vountas[1], John P. Burrows[1]**

[1] Institute of Environmental Physics, University of Bremen, Germany

[2]Leipziger Institut für Meteorologie, University of Leipzig, Germany

[3]School of Geospatial Engineering and Science, Sun Yat-Sen University, Zhuhai, P.R. China, 519082

## Abstract

To evaluate the performance of eXtensible Bremen Aerosol/cloud and surfacE parameters Retrieval (XBAER) algorithm, presented in part 1 of the companion paper, this manuscript applies the XBAER algorithm on the Sea and Land Surface Temperature Radiometer (SLSTR) instrument onboard Sentinel-3 and evaluates its performance. Snow properties: Snow Grain Size (SGS), Snow Particle Shape (SPS), and Specific Surface Area (SSA) are derived under cloud-free conditions. XBAER derived snow properties are compared to other existing satellite products and validated by ground-based/aircraft measurements. The atmospheric correction is performed on SLSTR for cloud-free scenarios using Modern-Era Retrospective Analysis for Research and Applications (MERRA) Aerosol Optical Thickness (AOT) and aerosol typing strategy according to the standard XBAER algorithm. The optimal SGS and SPS are estimated iteratively utilizing a Look-Up-Table (LUT) approach, minimizing the difference between SLSTR-observed and SCIATRAN simulated surface directional reflectances at 0.55 and 1.6 μm. The SSA is derived for a retrieved SGS and SPS pair. XBAER derived SGS, SPS and SSA have been validated using *in-situ* measurements from the recent campaign SnowEx17 during February 2017. The comparison shows a relative difference between XBAER-derived SGS and SnowEx17 measured SGS of less than 4%. The difference between XBAER-derived SSA and SnowEx17 measured SSA is 2.7 m$^2$/kg. XBAER-derived SPS can be reasonable-explained by

the SnowEx17 observed snow particle shapes. An intensive validation shows that (1) For SGS and SSA, XBAER derived results show high correlation with field-based measurements, with correlation coefficients higher than 0.85. The Root Mean Square Error (RMSE) of SGS and SSA are around 12 μm and 6 $m^2$/kg ; 2) For SPS, aggregate SPS retrieved by XBAER algorithm is likely to be matched with rounded grains while single SPS in XBAER is possibly linked to faceted crystals.

The comparison with aircraft measurements, during the Polar Airborne Measurements and Arctic Regional Climate Model Simulation Project (PAMARCMiP) campaign held in March 2018, also shows good agreement (with R=0.82 and R=0.81 for SGS and SSA, respectively). XBAER-derived SGS and SSA reveal the variability of the aircraft track of PAMARCMiP campaign. The comparison between XBAER-derived SGS results and MODIS Snow-Covered Area and Grain size (MODSCAG) product over Greenland shows similar spatial distributions. The geographic distribution of XBAER-derived SPS over Greenland and the whole Arctic can be reasonable-explained by campaign-based and laboratory investigations, indicating reasonable retrieval accuracy of the retrieved SPS. The geographic variabilities of XBAER-derived SGS and SSA over both Greenland and Arctic-wide agree with the snow metamorphism process.

## 1 Introduction

Change of snow properties is both a consequence and a driver of climate change (Barnett et al., 2005). Snow cover and snow season, especially in Northern Hemisphere, are reported by different models, to decrease due to climate change (Liston and Hiemstra, 2011). The reduction of snow cover leads to the change of surface energy budget (Cohen and Rind, 1991; Henderson et al.,2018), a reduction of Asian summer rainfall (Liu and Yanai, 2002; Zhang et al., 2019), a loss of Arctic plant species (Phoenix, 2018) and other impacts on societies and ecosystems (Bokhorst et al., 2016). Snow may influence the climate through both direct and indirect feedbacks (Lemke et al., 2007). The direct feedback is the snow-albedo feedback and the indirect feedbacks are involved by atmospheric circulation. The snow-albedo feedback

describes the mechanism that melting snow (the absence of snow cover), caused by global
warming, reflects less solar radiation, and further enhances the warming (Thackeray and
Fletcher, 2016). The snow indirect feedbacks describe the impact of snow properties change on
monsoonal and annual atmospheric circulation (Lemke et al., 2007; Gastineau et al., 2017).
However, the snow cover may be declining even faster than thought due to large uncertainties
of how models describe the snow feedback mechanisms (Flanner et al., 2011). The uncertainties
to describe the snow feedback mechanisms are largely introduced by the uncertainties of
knowledge of snow properties (Hansen et al., 1984; Groot Zwaaftink et al. 2011; Sarangi et al.,
2019). Snow properties depend on snow age, moisture, and surrounding temperatures
(LaChapelle,1969; Sokratov and Kazakov, 2012).
Model simulations and field-based measurements provide valuable information of snow
properties(e.g., Snow Grain Size (SGS), Snow Particle Shape (SPS), Specific Surface Area
(SSA)) for the understanding of changing snow and its corresponding impact on climate change.
Satellite observations offer another effective way to derive those snow properties on a large
scale with high quality (e.g. Painter et al., 2003; 2009; Stamnes et al., 2007; Lyapustin et al.,
2009; Wiebe et al., 2013). The similarities and differences of the required snow parameters and
their accuracy between the snow remote sensing community and other communities (e.g. field-
measurement community) are discussed in detail in part 1 of the companion paper (Mei et al.,
2020d).. In this manuscript, SGS (effective radius) is defined as $3V/(4A_p)$, where V and
$A_p$ are the volume and average projected area, respectively.
Different retrieval algorithms to derive SGS have been developed for different instruments.
Airborne Visible / Infrared Imaging Spectrometer (AVIRIS) and Thematic Mapper (TM)
onboard Landsat are pioneer instruments used for the retrieval of SGS (Hyvarinen and
Lammasniemi,1987; Li et al., 2001). Painter et al. (2003,2009) retrieved SGS using AVIRIS
and Moderate Resolution Imaging Spectroradiometer (MODIS) data, exploring the information
from both visible and near-infrared spectral channels. There are several available satellite SGS
products for MODIS (Klein and Stroeve, 2002; Painter et al., 2009; Rittger et al., 2013) and its
successor, Visible Infrared Imaging Radiometer Suite (VIIRS) (Key et al., 2013). For instance,
the MODIS Snow-Covered Area and Grain size (MODSCAG) product is created utilizing a

spectral mixture analysis method based on prescribed endmember. The endmember is a spectrum library for snow, vegetation, rock, and soil (Painter et al., 2009). The MODSCAG algorithm can provide snow cover fraction and snow albedo besides SGS on a pixel base. Topographic effects in MODSCAG are not considered and the MODSCAG product tends to overestimate SGS (Mary et al., 2013). Other retrieval algorithms have also been designed for and tested on the MODIS instrument (Stamnes et al., 2007; Aoki et al., 2007; Hori et al., 2007). Jin et al. (2008) retrieved SGS over the Antarctic continent using MODIS data based on an atmosphere-snow coupling radiative transfer model. Lyapustin et al. (2009) proposed a fast retrieval algorithm for SGS at a 1 km spatial resolution using MODIS observations. The algorithm is based on an analytical asymptotic radiative transfer model. Negi and Kokhanovsky (2011) proposed the use of the Asymptotic Radiative Transfer (ART) theory to retrieve SGS. The retrieved snow albedo and grain size from Negi and Kokhanovsky (2011) were validated and showed good accuracy for clean and dry snow. However, potential problems have been reported for dirty snow (e.g., soot/dust contamination). The Snow Grain Size and Pollution (SGSP) algorithm retrieves SGS and pollution amount based on a snow model (Zege et al., 1998), without a-priori assumptions on SPS (Zege et al., 2011). The SGSP algorithm has been validated using in-situ measurements over central Antarctica, and an underestimation of SGSP-derived SGS was reported under a large solar zenith angle (Zege et al., 2011; Carlsen et al., 2017). The algorithm is currently implemented for the MODIS instrument and provides operational daily snow products (Wiebe et al., 2011). New instruments such as Earth Observing-1 (EO-1) Hyperion and OLCI have also been used to derive SGS (Zhao et al., 2013; Kokhanovsky et al., 2019). The algorithm proposed by Kokhanovsky et al. (2019) is conceptually based on an analytical ART model, which estimates snow reflectance by given SGS and ice absorption (Kokhaovksy et al., 2018). The snow grains in the ART model are described as a fractal.

Snow particle shape is a fundamental parameter needed to describe snow properties (Räisänen et al. 2017). The SPS keeps relatively stable before falling on the ground under cold and dry conditions while it has large variabilities under warm and wet conditions (Dang et al., 2016). The International Classification for Seasonal Snow on the Ground (ICSSG) has

grouped the SPS into nine main morphological shapes: Precipitation Particles (PP), Machine
Made snow (MM), Decomposing and Fragmented precipitation particles (DF), Rounded Grains
(RG), Faceted Crystals (FC), Depth Hoar (DH), Surface Hoar (SH), Melt Forms (MF), Ice
Formations (IF) (Fierz et al., 2009). Another classification system, named as "global
classification" has been proposed in Nakaya and Sekido (1938) and has been updated recently
by Kikuchi et al. (2013). The "global classification" is obtained based on the SPS. The
information in Kikuchi et al. (2013) is qualitatively used to understand the satellite derived SPS
in this manuscript. Due to the complexity of the ice crystal shape, simplified ice crystal shapes,
such as fractal (Macke et al., 1996; Kokhanovsky et al., 2019) and droxtal (Pirazzini et al.,
2015), have been used in some satellite retrievals and model simulations. However, previous
investigations show that non-fractal snow types occur more frequently in reality (Gordon and
Taylo, 2009; Comola et al., 2017). Information of SPS, even limited or inaccurate, is extremly
helpufl and urgently needed for a better understanding of different snow types (Picard et al.,
2009). The widely used spherical shape assumption in field-based measurements (e.g., Flanner
and Zender, 2006) is not optimal for satellite-orientated retrievals, because the spherical shape
assumption can not produce the angular distribution of snow reflectance with required accuracy
(Leroux and Fily et al., 1998; Jin et al., 2008; Dumont et al., 2010; Mei et al., 2021), which will
introduce an unacceptable magnitude of uncertainty in the satellite retrieved snow properties.
Some attempts to derive ice crystal shape in ice clouds can be found in previous publications
(McFarlane et al., 2005; Cole et al., 2014). However, there is no publication with respect to the
retrieval of ice crystal shape in the snow layer using passive multi-spectrum satellite
observations. Although habit mixture models are preferable for the description of snow grain
shapes (Saito et al., 2019; Tanikawa et al., 2020; Pohl et al., 2020), the information content
from satellite observation is limited compared to field-based measurements. Thus, an optimal
single shape, which provides the best agreement between simulation and    satellite observation
(e.g. Top of the Atmosphere (TOA) reflectance) is also needed.
A few attempts have been proposed to retrieve SSA from space-borne observations. The
retrieval of SSA is actually performed based on the pre-retrieved SGS with an assumption of a
given known SPS. Mary et al. (2013) retrieved SSA over mountain regions using MODIS data,

assuming a spherical ice crystal shape. The algorithm performs a topographic correction for the surface reflectance to achieve a better retrieval accuracy. The overall difference, compared to field measurements, is 9.4 $m^2$/kg. Xiong et al. (2018) retrieved SSA using a snow reflectance model. The model simulates the light scattering process using a Monte Carlo method and shows an improvement of bidirectional reflectance, thus a better retrieval accuracy of SSA, compared to the spherical assumption. The overall difference, compared to field measurements, is about 6 $m^2$/kg.

This paper, as the companion paper of part 1, applies the XBAER algorithm on Sea and Land Surface Temperature Radiometer (SLSTR) onboard Sentinel-3 to derive SGS, SPS and SSA. The general concept is to use the channels, which are sensitive to SGS and SPS, simultaneously. The channels used in XBAER algorithms are 0.55 μm and 1.6 μm. An optimal SGS and SPS pair is achieved by minimizing the difference of atmospheric-corrected directional surface reflectances between satellite observations and SCIATRAN simulations. SSA is then calculated based on the retrieved SGS and SPS. Nine predefined ice crystal particle shapes (aggregate of 8 columns, droxtal, hollow bullet rosette, hollow column, plate, aggregate of 5 plates, aggregate of 10 plates, solid bullet rosette, column) (Yang et al., 2013) are used to describe the snow optical properties and to simulate the snow surface reflectance at 0.55 and 1.6 μm.

As mentioned in part 1 of the companion paper, the nine Yang SPSs used in the XBAER algorithm is proven to be a new option to describe the ice crystal local optical properteis for the snow community (e.g Saito et al., 2019; Pohl et al., 2020; Mei et al., 2021), we would also like to emphasize several more point to avoid misunderstandings between different scientific communities.

> **Difference between field-measured and satellite-derived SPS**. A field-measured SPS is an optical shape for a single ice crystal while satellite-derived SPS is an averaged radiative shape on a certain geographic area. The geographic area is determined by the instrument spatial resolution (1 kilometer as used in this study). Thus it is unreasonable to directly compare a kilometer average radiative shape to a

single ice crystal shape. However, for a region with a similar snow metamorphism process (Colbeck et al., 1980;1983), the field measured SPS may provide some representative information with respect to if the ice crystal shape is convex (e.g. spherical shape) or non-convex (aggregate shape), which is also critical for further applications. This fundamental difference between field-measured and satellite-derived SPS restricts that only a qualitative evaluation of the satellite retrieved SPS is possible. Please be noted that this spatial resolution issue is more than just a typical "general scale issue" becuase it fully depends on the paramters retreived, especially on their inhomogeneity.

➤ **Requests to describe snow properties in the radiative transfer theory**: there is another way to describe snow properties in the radiative transfer theory. This manner needs no knowledge with respect to SPS, but use an assumption of stochastic medium. However, in this manner, there are also parameters (e.g. mean photon path length) which cannot be validated. It is worth to notice that, all manners, for the retrieval of snow properties from satellite, needs to make some assumptions. These assumptions are fundamentally needed for a specific retrieval algorithm (Langlois et al., 2020).

➤ **Different radiative transfer models used for snow community**: For the widely used Asymptotic radiative transfer (ART) model, even though the users do not highlight the issues linked to SPS, these issues exist. (1) The original ART model (Zege et al., 2004; Kokhanovsky and Zege et al., 2005) is derived based on the assumption of second-generation fractal for ice crystal shape; (2) In the updated ART model (Kokhnaovsky et al 2018), $g$ and $B$ parameters are introduced. The $g$ parameter depends on both SGS and SPS. The $B$ parameter depends strongly on SPS (Libois et al., 2014). Even one can state that the $g$ and $B$ parameters can be fitted to real observations, several issues linked to the assumption of SPS occur: (1) the accuracy of use a single $g$ parameter to describe the complicated particle phase function needs to be checked; (2) ART model is designed for medium with weakly absorption properties, thus it cannot be used for certain SGS and SPS, especially for long wavelength (e.g. 1.6 µm). In short, we cannot really avoid making certain (explicit or

hidden) assumptions of SPS if it is not iteratively retrieved in the algorithm, like in

the eXtensible Bremen Aerosol/cloud and surfacE parameters Retrieval (XBAER)

algorithm.

➢ **Highlighting with respect to the XBAER retrieved SPS**: We believe our work, as

a first step/attempt, provides some new/useful way/information for the SPS. However,

we should not over-interpret the shape we retrieved.

This paper is structured as follows: instrument characteristics of SLSTR and the field-
based measurements and aircraft measurements used for validation are described in section 2.
Section 3 describes the method including cloud screening, atmospheric correction, and the
flowchart of the XBAER algorithm. Some selected data products and comparisons with
MODIS products and field-based measurements are shown in section 4. The comparison with
the recent campaign measurement is presented in section 5. A discussion to illustrate a time
series of the retrieval results is shown in section 6. The conclusions are given in section 7.

**2 Data**
**2.1 SLSTR instrument**
After the loss of Environmental Satellite (Envisat) on 12 April 2012, the European Space
Agency (ESA) launched Sentinel-3A, Sentinel-3B in February 2016, and April 2018,
respectively. As the successor of Advanced Along-Track Scanning Radiometer (AATSR)
onboard Envisat, Sentinel satellites take the SLSTR instrument. The SLSTR instrument has
similar characteristics as compared to AATSR (see Table 1 for details). The instrument has
nine spectral bands in the visible and infrared spectral range. It also has dual-view observation
capability with swath widths of 1420 km and 750 km for nadir and oblique directions,
respectively. The SLSTR and AATSR dual-view observations of the Earth's surface make
surface Bidirectional Reflectance Distribution Function (BRDF) effect estimation possible,
which is widely used to retrieve both surface and atmospheric geophysical parameters (Popp et
al., 2016). Besides the heritage of AATSR, some new features (wider swath, new spectral bands
and higher spectral resolution for certain bands) have been included in SLSTR instrument
(https://sentinel.esa.int/web/sentinel/technical-guides/sentinel-3-slstr/instrument).

Table 1 Instrument characteristics of AATSR and SLSTR

| SLSTR | | | AATSR | | |
|---|---|---|---|---|---|
| Band # | Central wavelength(µm) | Resolution(m) | Band # | Central wavelength(µm) | Resolution(m) |
| 1 | 0.555 | 500 | 4 | 0.555 | 1000 |
| 2 | 0.659 | 500 | 5 | 0.659 | 1000 |
| 3 | 0.865 | 500 | 6 | 0.865 | 1000 |
| 4 | 1.375 | 500 | | | |
| 5 | 1.610 | 500 | 7 | 1.610 | 1000 |
| 6 | 2.25 | 500 | | | |
| 7 | 3.74 | 1000 | 1 | 3.74 | 1000 |
| 8 | 10.85 | 1000 | 2 | 10.85 | 1000 |
| 9 | 12 | 1000 | 3 | 12 | 1000 |
| 10 | 3.74 | 1000 | | | |
| 11 | 10.85 | 1000 | | | |


## 2.2 Ground-based measurements


The validation of satellite derived snow properties is challenging due to i) limited available
field-based measurements; ii) the difficulties of spatial-temporal collocation between satellite
observations and field-based measurements because of cloud coverage. This manuscript
focuses on the Sentinel-3a satellite for the period of February 2016 (lauch month of Sentinel-
3a) and December 2020. The field-based measurements from both permanent sites and
campaign sites for the focusing time period are collected. Fig. 1 shows the geographic
distribution of the validation sites. The site names used in this manuscript are listed near each
site. Since XBAER retrieves SGS, SPS and SSA simultaneously, the SnowEx campaign, which
provides three parameters as well, will be introduced detailed first.

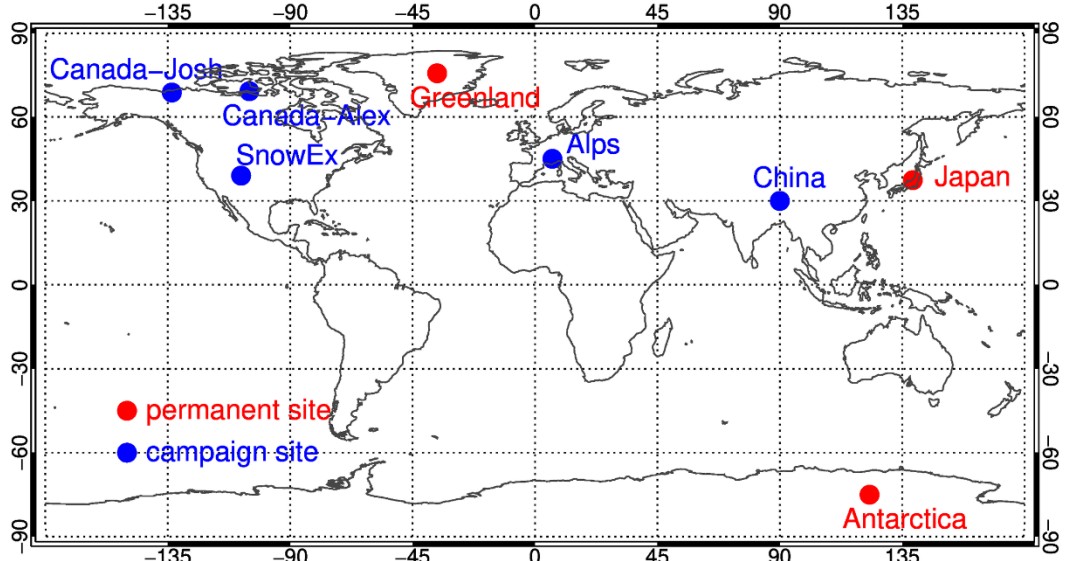

Fig. 1 Geographic distribution of the validation sites. The colors represent the type of each site while the site name used in this manuscript is indicated near each site.

NASA established a terrestrial hydrology program (SnowEx mission) in order to better quantify the amount of water stored in snow-covered regions (Kim et al., 2017). The measurements for the first year (2016 - 2017) were carried out during February 2017 (between 08 February 2017 and 25 February 2017) at Grand Mesa and the Senator Beck Basin in Colorado (hereafter refer as SnowEx17) (See Fig. 2 (a)) (Elder et al., 2018). Grand Mesa is a forest region covered by relatively homogeneous snow cover with an area size similar to airborne instrument swath widths (Brucker et al., 2017) (See Fig. 2 (c)). Senator Beck Basin site has a complex topography and covered by snow. The campaign used more than 30 remote sensing instruments and most of the instruments are from the National Aeronautics and Space Administration (NASA) except some instruments such as the ESA's Radar (Kim et al., 2017). The snowpits measurements provide information of snow grain size and type/shape, stratigraphy profiles, and temperatures with certain information about surface conditions (e.g. snow roughness) (Rutter et al., 2018). The SnowEx17 campaign provides seven different shapes (New Snow, Rounds, Facets, Mixed Forms, Melt-Freeze, Crust, and Ice Lens). Table 2 lists both the SnowEx17 measured snow grain shapes and SPSs defined in Yang et al. (2013). The SPSs defined by ICSSG are also listed in the table and the possible linkage between Yang

SPS and ICSSG SPS (named as SPS similarity) will be discussed later. The measurements have
been publicly released in nsidc.org/data/snowex. The data was collectd in SnowEx20 for the
period of 27 January and 12 February, 2020.


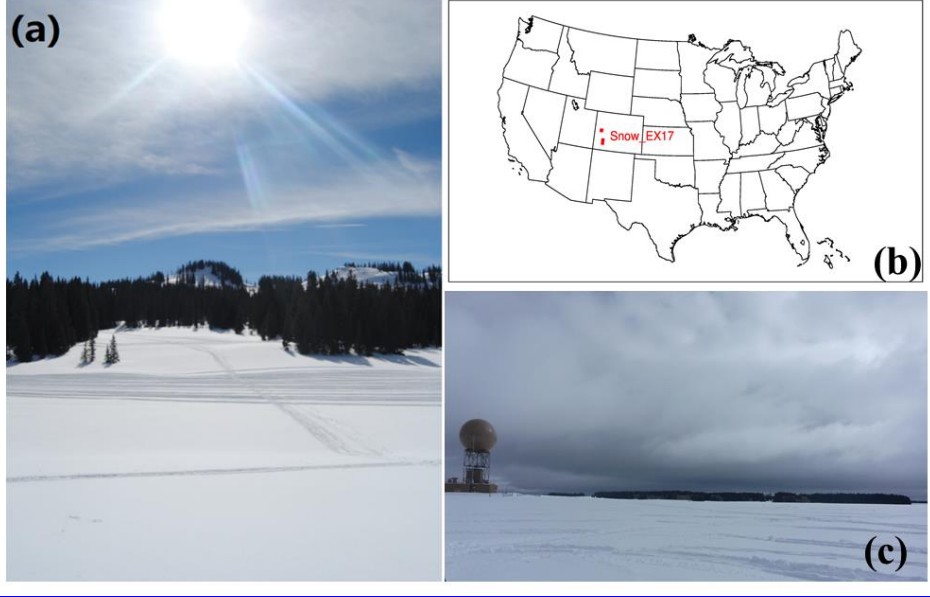



Fig. 2 Photos taken during the SnowEx campaign. (a) An overview of the campaign
environment around Senator Beck Basin site; (b) Location of SnowEx campaign (red
rectangles); (a) An overview of the campaign environment around Grand Mesa site;. Roy, A.
Langlois, and L. Brucker; supplied by the National Snow and Ice Data Center, University of
Colorado, Boulder)
The measurements over Greenalnd are obtained by the EastGRIP team over
(75.63°N,36.004°W) . Detailed information of the site can be found at https://eastgrip.org.
The data have been used to validate the SGS and SSA derived from OLCI (Kokhanvosky et
al., 2018). The same dataset, covering the period of May 2017 and August 2018 is used in this
manuscript.
The SSA measurements at Nunavut, Northern Canana (69.20°N,104.80°W) were
obtained using the instrument described by Montpetit et al. (2012). The observation period
covers April, 2018. SGS or SSA is calculated using the relationship between SSA and SGS if
SSA or SGS is not measured.
The SPS and SSA measurements around Inuvik, Northwest Territories of Canana
(68.73°N,133.49°W) covers the period of November 2018 – March 2019. There were three
deployments, the freeze-up period (November 2018), the strom input period (Janauray 2019)
and the metamorphosis period (March 2019) (King et al., 2019).
The SSA measurements above Frech Alps (45.04°N,6.41°W) were collected in the snow
seasons during 2016 – 2018 (Tuzet et al,, 2020). The measurements for 2016 – 2017 period
provide SSA profile information with vertical resolution of 3 cm using the DUFISSS
instrument (Gallet et al., 2009). For the period of 2017 -2018, the measurements were
obtained with vertical resolution of 6 cm using the Alpine Snowpack Specific Surface Area
Profiler (Libois et al., 2014). The uncertainty is estimated to be 10%.The SGS measuremnts
were obtained over Nagaoka, Japan (37.41°N,138.88°W) (Yamaguchi et al., 2019; Avanzi et
al., 2019). The observations during January, 2017 – March 2018 are used in this manuscript.
The SGS measurements were obtained over Xingjiang province during diferent period
(Chen et al., 2020), the dataset around the site (44.146°N,85.848°E) for the period November
2018 – November 2019 is used in this manuscrip.
The SSA measurements at Dome C (75°S,123°E) in Antarctica cover the period of 2016
– 2018, the accracy of the measurments is better than 15% (Picard et al., 2016). The data were
collected using a self-designed and assembled insturment, named as Autosolexs, which can be
used to measure the snow properties for several years under the harsh environment.




Table 2 Snow grain type (shape) provided by Yang et al (2013) , *in-situ* measurements in
SnowEx campaign and by ICSSG. Please note here the grain type by Yang et al.,measured in

SnowEx and provided by ICSSG given in the same line have no 1:1linkage


| Yang | | | SnowEx | ICSSG | |
|------|-----------|-------------------|---------------|--------------------------|-----------|
| **Grain Type** | **Abbriation** | **Schematic drawing** | **Grain Type** | **Grain Type** | **Abbriation** |
| Aggregate of 8 columns | col8e | | New Snow | Precipitation Particles | PP |
| Droxtal | droxa | | Rounds | Machine Made snow | MM |
| Hollow bullet rosettes | holbr | | Facets | Decomposing and Fragmented | DF |
| Hollow column , | holco | | Mixed Forms | Rounded Grains | RG |
| Plate | pla_1 | | Melt-Freeze | Faceted Crystals | FC |
| Aggregate of 5 plates | pla_5 | | Crust | Depth Hoar | DH |
| Aggregate of 10 plates | pla_10 | | Ice Lens | Surface Hoar | SH |
| Solid bullet rosettes | solbr | | - | Melt Forms | MF |
| Column | solco | | - | Ice Formations | IF |

## 2.3 Aircraft observations

During the Polar Airborne Measurements and Arctic Regional Climate Model Simulation
Project (PAMARCMiP) campaign held in March/April 2018 ground-based and airborne
observations of surface, cloud and aerosol properties were performed near the Villum Research
Station (North Greenland). One of the most important objectives of the PAMARCMiP 2018
campaign was to quantify the physical and optical properties of snow, sea ice and atmosphere
(Egerer, et al., 2019; Nakoudi et al., 2020). Airborne spectral irradiance measurements by the
Spectral Modular Airborne Radiation Measurement System (SMART) onboard the Polar 5
research aircraft operated by Alfred-Wegener-Institut were used to derive snow grain sizes
along the flight track. The SMART provides solar up- and downward spectral irradiances in the
range between $0.4 - 2.0\,\mu m$. The optical inlets are actively horizontally stabilized with respect
to aircraft movement (Wendisch et al., 2001) within 5° pitch and roll angle. In particular, for
high solar zenith angles (SZA) as presented during PAMARCMiP (about 80° SZA),
misalignment of the optical inlets implies significant measurement uncertainties (Wendisch et
al., 2001). Further uncertainties are related to the spectral and radiometric calibration, as well
as the correction of the cosine response which sums up to a total wavelength-dependent
uncertainty (one sigma) for the irradiances ranging between 3 to 14% (Jäkel et al., 2015). The
derivation of the surface albedo from aircraft observations requires atmospheric corrections due
to the atmospheric attenuation and scattering by gases and aerosols. There an iterative method
to correct for these effects was applied according to the procedure described by Wendisch et al.
(2004). The retrieval of the snow grain sizes is based on the method described in Carlsen et al.
(2017) which uses a modified approach presented by Zege et al. (2011).
**3 Methodology**
**3.1 Cloud screening**
The algorithm synergistically uses SLSTR and OLCI data to identify clouds over the snow
surface. The criteria for cloud screening over snow using SLSTR and OLCI measurements can
be found in Istomina et al. (2010) and Mei et al. (2017), respectively. Short summaries of
Istomina et al. (2010) and Mei et al. (2017) are presented below and more details can be found
in the original publications. The algorithm proposed by Istomina et al. (2010) for the SLSTR
instrument utilizes spectral behavior differences at SLSTR visible and thermal infrared
channels, and this algorithm is updated later by Jafariserajehlou et al. (2019). Relative
thresholds are determined based on radiative transfer simulations under various atmospheric
and surface conditions. The method proposed by Mei et al. (2017) for the OLCI instrument uses
different cloud characteristics: cloud brightness, cloud height, and cloud homogeneity. The
TOA reflectance at $0.412\,\mu m$, the ratio of TOA reflectance at 0.76 and $0.753\,\mu m$, standard
deviation of TOA reflectance at $0.412\,\mu m$ are used to characterize cloud brightness, cloud

height, and cloud homogeneity, respectively. A pixel is identified as a cloud-free snow pixel when both SLSTR and OLCI identify it as a cloud-free snow pixel. Identified clouds can be surrounded by a so-called "twilight zone" (Koren et al., 2007), which can extend more than ten kilometers from a cloud pixel to a cloud-free area. The surrounding 5×5 pixels of an identified cloud pixel will be marked as a cloud to avoid the "twilight zone" effect. A more detailed description of this cloud screening method can be found in Mei et al. (2020a). Addtionally, TOA reflectance at 0.55 μm is required to be higher than 0.5 to avoid dark ice and dirty snow.

## 3.2 Atmospheric correction

Due to the low atmospheric aerosol loading over the Arctic snow covered regions (e.g. Greenland), atmospheric correction using path radiance representation (Chandrasekhar, 1950; Kaufman et al., 1997) can provide accurate estimation of surface reflection even under relatively large SZA (Lyapustin, 1999). The TOA reflectance at selected channels (0.55 and 1.6 μm) is described by the path radiance representation (Chandrasekhar, 1950; Kaufman et al., 1997) as:

$$R(\theta,\theta_0,\varphi,\tau,AT) = R^0(\theta,\theta_0,\varphi,\tau,AT) + \frac{T(\theta,\theta_0,\tau,AT)A}{1-s(\tau,AT)A}, \tag{1}$$

where $R^0(\theta,\theta_0,\varphi,\tau,AT)$ is the TOA reflectance calculated assuming black surface (surface reflectance equal 0) under VZA, SZA and RAA of $\theta,\theta_0,\varphi$. $\tau$ and AT are AOT and aerosol type. $T(\theta,\theta_0,\tau,AT)$ is the total (diffuse and direct) transmittance from the sun to the surface and from surface to the satellite, $s(\tau,AT)$ is spherical albedo, $A$ is Lambertian surface albedo. The spherical albedo is the fraction of the incident solar radiation diffusely reflected over all directions (albedo of an entire planet). The Lambertian surface albedo is defined as the ratio of reflected to incident flux. The atmospheric correction is performed based on the following equation:

$$A = \frac{R(\theta,\theta_0,\varphi,\tau,AT) - R^0(\theta,\theta_0,\varphi,\tau,AT)}{(R(\theta,\theta_0,\varphi,\tau,AT) - R^0(\theta,\theta_0,\varphi,\tau,AT))s(\tau,AT) + T(\theta,\theta_0,\tau,AT)}. \tag{2}$$

The atmospheric correction is based on the Look-Up-Table (LUT) precalculated using
radiative transfer code SCIATRAN (Rozanov et al., 2014). The radiative transfer calculations
were performed assuming AOT values provided by MERRA simulations and aerosol type
defined as weakly absorbing according to a previous investigation (Mei et al., 2020b).

**3.3 XBAER Algorithm**
The theoretical background of the retrieval algorithm is given in section 4 of the companion
paper. The XBAER algorithm consists of three stages to derive SGS, SPS, and SSA: 1)
derivation of SGSs for each predefined SPS; 2) selection of the optimal SGS and SPS pairs
for each scenario; 3) calculation of SSA for each retrieved SGS and SPS. This section
describes some implementation details such as the selection of the first guess for the retrieval
parameters and the flowchart of the algorithm.
A reasonable first guess value for the iteration process can significantly reduce the
computation time, which is important for retrievals of atmospheric and surface properties over
large geographic and temporal scales with different instrument spatial resolutions. The first
guess of SGS in the XBAER algorithm is obtained employing the semi-analytical snow
reflectance model (Kokhanovsky and Zege, 2004; Kokhanovsky et al., 2018). Details of using
this model to derive SGS can be found in Lyapustin et al. (2009). Due to the different band
settings in MODIS and SLSTR (SLSTR has no 2.1 μm channel as MODIS), one non-absorption
channel (0.55 μm) and one absorption channel (1.6 μm) are used in our SLSTR retrieval
algorithm.
Fig. 3 shows the flowchart of how XBEAR derives SGS, SPS, and SSA. The flowchart
includes pre-processing of cloud screening using the synergy of OLCI and SLSTR and the
atmospheric correction using MERRA providing AOT and weakly absorbing aerosol type. The
SGS and SPS are obtained using the LUT-based minimization routine. SSA is then calculated
using the retrieved SGS and SPS.

**XBAER: eXtensible Bremen Aerosol/cloud and surfacE parameters Retrieval (snow)**

XBAER input for SLSTR nadir observations
- SLSTR TOA reflectance at 0.55, 0.67, 0.87, 1.6 μm
- Sun Zenith Angle($\mu_0$), Viewing Zenith Angle($\mu$), Relative Azimuth Angle($\varphi$)
- Longitude, Latitude, Time

Identify bright pixels, mask out dark pixels and fill the values
Cloud Mask with the following criteria(P1,P2 and P3 are threshold values)
- $(R_3-R_4)/R_3 > P1$ and $(R_3-R_2)/R_3 < P2$ and $(R_2-R_1)/R_2 < P3$
- 5×5 pixel to remove cloud adjacency effect
XBAER_standard cloud screening

OLCI TOA reflectance
- 0.412 μm, 0.756 μm , 0.76 μm

**Atmospheric correction**
- AOT (τ)
- Aerosol Type (weakly absorbing)

No retrieval ← **NO** — Cloud free **snow**

**First iteration**
- SGS first guess($r_0$)

**Yes**

For 0.55 and 1.6 μm, interpolate LUT on geometry and SGS
**Snow surface reflectance estimation** based on Eq. (3) for a given ice crystal shape.

For 0.55 and 1.6 μm, interpolate LUT on geometry and AOT
**Atmospheric Correction** based on Eq. (2) for a given AOT(τ) and aerosol type

**Find SGS and SPS such that** $\|A_e(r_i,SPS) - R_s(r_i,SPS)\| \to min$

**NO**

**EXIT** ← **Yes** — Max iteration — **NO** → Abs($r_i$-$r_{i-1}$)<0.1% — **Yes** → XBAER output
- **SGS**
- **SPS** → **SSA**

$r_i$ - SGS for iteration step i    $r_{i-1}$ - SGS for iteration step i-1


Fig. 3 Flow chart of the XBAER retrieval algorithm


## 4 Results and Comparison

Greenland is the largest ice-covered land mass in the northern hemisphere and the biggest cryospheric contributor to the global sea-level rise (Ryan et al., 2019). XBAER derived SGS, SPS, and SSA over Greenland enable a good understanding of the retrieval accuracy with a large and representative geographic scale. Kokhanovsky et al., (2019) reported that July is an optimal month to analyze satellite-derived snow properties over Greenland because Greenland has a strong Snow Particle Metamorphism Process (SPMP) due to higher temperatures in July (Nakamura et al. 2001). The SPMP, affected strongly by temperature, is a dominant factor for the variabilities of SGS, SPS, and SSA (LaChapelle,1969; Sokratov and Kazakov, 2012; Saito et al., 2019). Snow particle size increases dramatically and the ice crystal particles are compacted in the strong SPMP (Aoki et al., 1999; Nakamura et al. 2001; Ishimoto et al. 2018).

Fig. 4 shows an example of the XBAER-derived SGS on 28 July 2017 from SLSTR, XBAER first guess and its comparison with the same scenario from MODSCAG product (Painter et al., 2009). Here we chose MODIS/Aqua rather than MODIS/Terra to avoid the impact of instrument degradation of MODIS/Terra (Lyapustin et al., 2014). The visualization of XBAER-derived SGS is shown to be between 10 and 500 μm. The XBAER first guess has in general low value (Lyapustin et al., 2009), as compared to XBAER and MODSCAG results, . The XBAER and MODSCAG derived SGS show good agreement on the geographic distribution. The slight difference of cloud covered regions (white parts) is explained by the different overpass time between SLSTR and MODIS. Both algorithms demonstrate that SGSs in central Greenland are smaller than those at coastline regions. This is attributed to the geographic distribution of surface temperature over Greenland. In particular, central Greenland has a significantly higher elevation and the impacts of imperfect atmospheric correction on retrieved snow properties are ignorable. The lower temperature under higher elevation regions has weaker SPMP, producing more irregular SPS. The situation is opposite in the coastline regions over Greenland. Since Fig. 4 is composed by three different SLSTR orbits, the geometrical-shaped features in Eastern Greenland are caused by the effective Lambertian albedo assumption in XBAER algorithm. This assumption introduces addtional bias under large viewing zenith angle condition, which occurs at the edge of each SLSTR orbit.

Fig. 5 shows XBAER retrieved SGS, SPS, and SSA for 28 July 2017. Since there are no
available products of SPS and SSA from MODSCAG, it is a great challenge to do a similar
comparison as in the case of SGS. Fortunately, campaign-based and laboratory investigations
provide valuable information on typical snow shapes under different times/locations with a
wide range of atmospheric conditions. According to Kikuchi et al. (2013), the typical SPSs in
the polar regions include column crystal (e.g. solid column, bullet-type crystal) with SGS of
about 50 μm for solid column and between 100 μm and 500 μm for bullet-type, the germ of ice
crystal group with SGS of less than 50 μm. Saito et al. (2019) pointed out that SPSs of fresh
snow in the polar regions are typically a mixture of irregular shapes such as column and
platelike shape. Ishimoto et al. (2018) found that aged snow can have an aggregate structure.
The optical properties of small ice crystal particles in aged snow may be well-characterized by
granular/roundish shapes, while SPS tends to be irregular or severely roughened shapes during
the SPMP (Ishimoto et al., 2018). Pirazzini et al (2015) investigated the impact of ice crystal
sphericity on the estimation of snow albedo and found droxtal is a reasonable assumption to
take ice particle non-sphericity into account. The above conclusions can be used as qualitative
reference to understand the satellite-derived SPS. In the meantime, a large proportion of ice-
sheet melts during the warm July, which unequivocally leads to rounded coarse grains very
quickly. According to Fig. 5, central Greenland is largely covered by small particles with
roundish/droxtal shape while coastline regions are covered to be aggregated shapes (aggregate
of 8 columns, aggregate of 5 plates, the aggregate of 10 plates) with large particle sizes, are
essentially attributed to the different SPMP over different regions of Greenland. Bullet-type
crystal (solid bullet rosettes) occurred with SGS of about 100 μm. The examples shown in Fig.
5 can be reasonably explained by previous publications (Kikuchi et al., 2013; Pirazzini et al.,
2015; Ishimoto et al.,2018; Saito et al., 2019).

The geographic distribution of SSA is somehow anti-correlated with the geographic

distribution of SGS, due to the definition of SSA. Most SSA fall into the range of 10-40 $m^2$/kg,
which agrees with previous publication (Kokhanovsk et al., 2019). The change of SSA occurs
especially after snowfall (Carlsen et al., 2017; Xiong et al., 2018). Since SSA contains both
information of SGS and SPS and field measurements provide SSA, the validation of SSA can
be also used as an „indirect quantitative validation" of SPS, which will be quantitatively
presented in the next section.

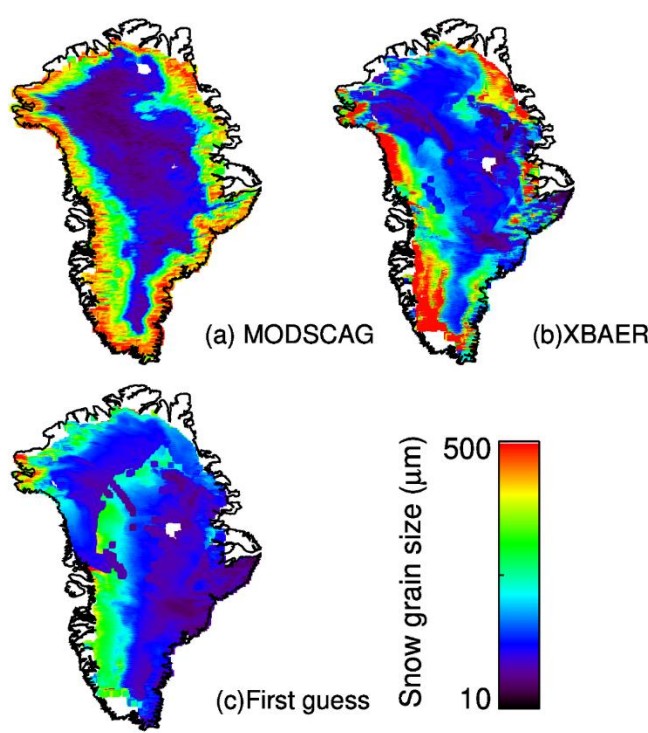


Fig 4. A comparison of the MODISSCAG SGS (a) ; XBAER derived SGS (b) and first guess
(c) over Greenland on 28 July, 2017.

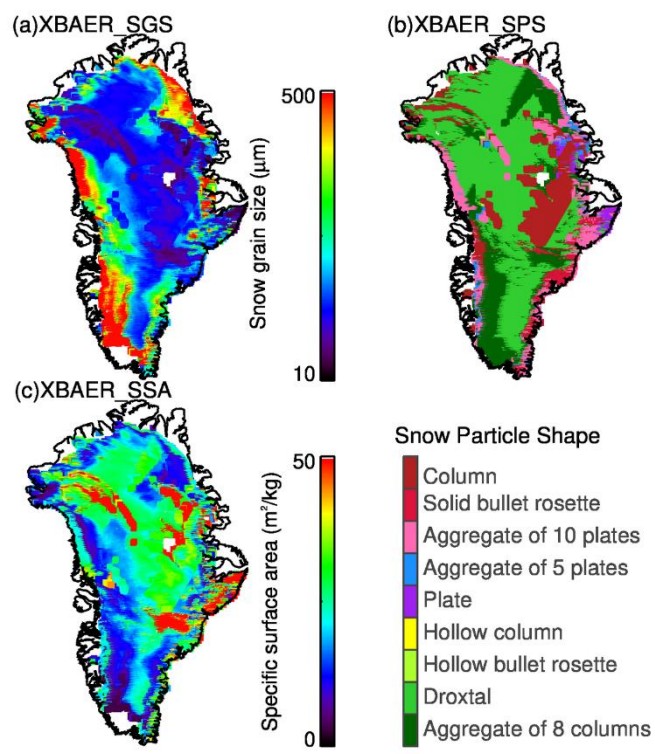


Fig 5. XBAER derived SGS, SPS and SSA over Greenland for the same scenario as in Fig. 4.

## 5 Validation

In this section, we will quantitatively validate XBAER derived snow properties with field-
based and aircraft measurements.

### 5.1 Validation using the observations of SnowEx17 campaign

In order to have a quantitative evaluation of XBAER-derived SGS, SPS, and SSA, we have
collocated the SLSTR observations with recent campaign measurements provided by
SnowEx17 and SnowEx20, as described in section 2. Due to overpass time and cloud cover,
only limited match-ups between XBAER retrievals and SnowEx17 and SnowEx20
measurements have been obtained. No match-up is obtained for SnowEx20.
Table 3 summarizes match-up information. The first three columns in Table 3 show the
observation times and locations (longitude and latitude). The fourth and fifth columns indicate
the cloud conditions. Cloud conditions in Table 3 are given by three categories: cloud-free snow,
cloud-contaminated snow, and cloud-covered snow. These three categories are classified by the
XBAER cloud identification results (see Section 3.1) and are illustrated by the RGB
composition figures, covering the SnowEx campaign area, as presented in Fig. 6. An optically
thin cloud over a melting snow layer, a thick cloud over snow, and snow scenarios are presented
in Fig. 6 (a), (b) and (c), respectively. The cloud optical thickness (COT), estimated using the
independent XBAER cloud retrieval algorithm, as presented in Mei et al (2018), is ~0.5 and
~10 for 9[th] and 11[th] February, respectively.
Table 3 Information of Match-ups between SnowEx and SLSTR during February, 2017

| Date | Lon(°) | Lat(°) | COT | Comment |
|---|---|---|---|---|
| 02-09 | -108.1092 | 39.0369 | ~0.5 | cloud-contaminated snow |
| 02-22 | -108.0634 | 39.0444 | 0 | cloud-free snow |
| 02-22 | -108.0625 | 39.0459 | 0 | cloud-free snow |
| 02-22 | -108.0617 | 39.047 | 0 | cloud-free snow |
| 02-11 | -108.0462 | 39.0278 | ~10 | cloud-covered snow |



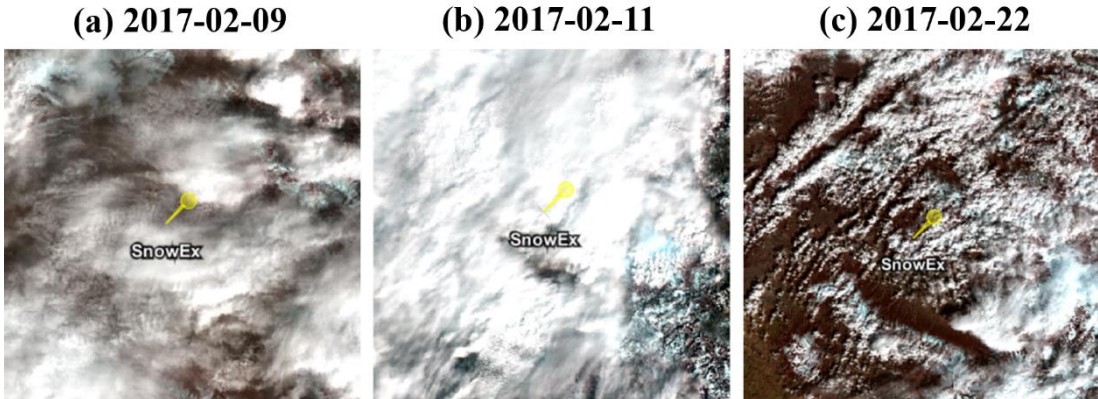

**(a) 2017-02-09**     **(b) 2017-02-11**     **(c) 2017-02-22**


Fig 6. Zoom-in of the RGB composition figures (created using ESA official SLSTR software
SNAP) for the selected 3 days presented in Table 3. The yellow point indicate the SnowEx
instrument position.

Even though the synergistical use of SLSTR and OLCI provides valuable information to
separate cloud and snow, the identification of an optically thin cloud above a snow layer is a
great challenge due to the similar wavelength dependence of snow and cloud reflectance,
especially between snow and ice cloud (Mei et al., 2020). The identification of the cloud from
an underlying snow layer in XBAER relies mainly on the $O_2$ channel on OLCI instrument,
which provides the cloud height information (Mei et al., 2017). Fig. 7 shows the performance

of XBAER cloud identification results for cloud contamination and cloud-covered snow scenarios. The red star indicates the measurement location. The zoom-in figures around the measurement site are presented in Fig. 6 above. XBAER cloud screening shows, in general, a good performance according to the RGB visual interpretation. However, part of the thin cirrus cloud on the 9[th] of February is not correctly avoided. For 9[th] of February, XBAER cloud identification gives a result of clean snow while it contains a thin cloud above a snow layer. For the 11[th] of February, XBAER has successfully detected the cloud from an underlaying snow layer. For a comprehensive investigation of XBAER derived snow properties under all snow-cloud coupled conditions, the match-up on 11[th] February 2017 (labeled as grey) has been manually set to be „cloud free snow". The reason to perform the validation for different cloud conditions is that the satellite retrieval can only be performed under cloud-free conditions while field measurements may be obtained under cloud conditions, especially when fresh snow properties are measured. Thus, the field-based measurements under full-cloud or partly-cloudy conditions are still valuable in the validation process (Jeoung et al., 2020). According to the sensitivity study, cloud contamination leads to an underestimation of SGS and the overestimation of SSA, depending on the cloud fraction.

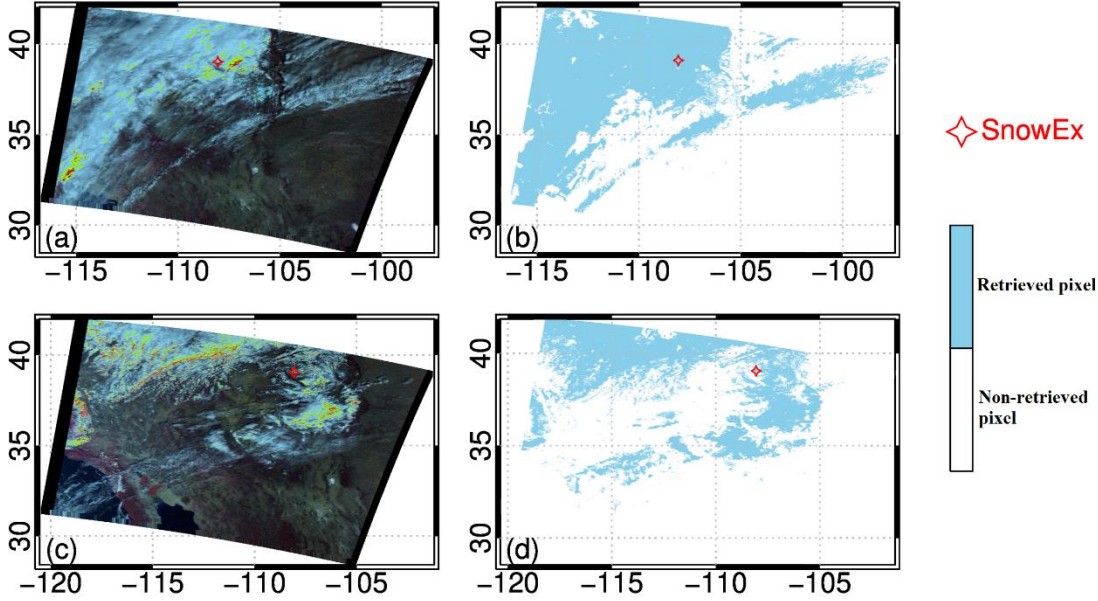

Fig 7. The RGB composition (left column) for 9 (a) and 22 (c) February when XBAER detect as cloud free snow and provides the retrieval. The XBAER cloud screening results (right column) for the corresponding days are given in (b) and (d). The "Retrieved pixel" legend refers

to cloud free snow. The "Non-retrieved pixel" legend refers to the area where XBAER retrieval
is not performed, this includes (1) snow-free and cloud free (2) cloud above snow; (3) cloud
above snow-free.

Table 4 summarizes the comparison between XBAER retrieval results, MODSCAG

product, and SnowEx17 campaign measurements. The first three columns in Table 4 are the
same as Table 3, showing the observation time and locations (longitude and latitude). The
second three columns are the SnowEx17 measured SGS. Since the SnowEx17 provides the
SGS profile up to 1 meter depth, the minimum (SnowEx_min), average (SnowEx_avg), and
maximum (SnowEx_max) values of SGS are listed in Table 3. The last two columns are
MODSCAG and XBAER derived SGS. For the four cloud-filter-passed match-ups, XBAER-
derived SGS shows good agreement with SnowEx17 measurements, especially for the 22[nd] of
February. The average absolute difference is less than 10 μm (4 % in relative difference). The
relatively large SGS ($\geqslant$ 250μm) caused mainly by the warm-up on the 21[st] of February (see
the comment in Table 5, reported by campaign participators), which leads to a quicker snow
metamorphism process, forming large ice crystal particles. MODSCAG only provides retrieval
results for 9[th] and 11[th] Feb. The results from XBAER and the MODSCAG agree well. This
possibly indicate a similar performance between XBAER and the MODSCAG.

An underestimation is found for the first match-up on the 9[th] of February. This is explained

by the cirrus cloud contamination as presented in Fig. 11. According to an independent XBAER
cloud retrieval (Mei et al., 2018), the COT is ~0.5, cloud contamination with COT=0.5
introduces ~30% underestimation according to fig. 11 in part 1 of the companion paper. So for
SGS=100 μm, provided by SnowEx, XBAER is expected to have a theoretically retrieved SGS
of ~ 70 μm while a value of 78.2 μm is obtained from the real satellite retrieval. In order to
further confirm this negative bias feature caused by cloud contamination, 11[th] February (a
snowstorm at the measurement site is reported by campaign participators), although filtered by
the XBAER cloud screening routine, is forced to retrieve the full-cloud-covered scenario as a
cloud-free case. According to the theoretical investigations presented in part 1 of the companion
paper, for COT≥5, the XBAER algorithm retrieves cloud effective radius, rather than SGS.
The retrieved ice crystal size depends on the cloud effective radius of the cloud above the
underlying snow layer. The independent XBAER cloud retrieval provides SGS value of ~ 38
μm while 32.3 μm is obtained by the XBAER snow retrieval, for a reference value of 100 μm
as provided by SnowEx17 measurement. This is consistent with a typical ice cloud effective
radius (King et al., 2013; Mei et al., 2018), under a snowstorm condition.

Table 4 The comparison between SnowEx SGS measurements,XBAER and MODSCAG
retrieved SGS during February, 2017.

| Date | Lon(°) | Lat(°) | SnowEx_ min(μm) | SnowEx_a vg(μm) | SnowEx_ max(μm) | MODSCAG (μm) | XBAER(μm) |
|------|--------|--------|------|------|------|---------|-----------|
| 02-09 | -108.1092 | 39.0369 | 50 | 100 | 150 | 90 | 78.2 |
| 02-11 | -108.0462 | 39.0278 | 50 | 100 | 200 | 40 | 32.3 |
| 02-22 | -108.0634 | 39.0444 | 100 | 250 | 500 | - | 254.4 |
| 02-22 | -108.0625 | 39.0459 | 150 | 250 | 400 | - | 254.4 |
| 02-22 | -108.0617 | 39.047 | 100 | 200 | 300 | - | 215.7 |


Table 5 shows the same match-up information as in Table 4, but for SPS. We would like
to highlight again, the SPSs proposed by Yang et al (2013) are used for the radiative transfer
calculation. From a single ice crystal point of view, those shapes are very unlikely to occur
exactly in reality. This is similar to the issue in field measurements. In field-based
measurements, spherical shape assumption is widely used (e.g., the calculation of SSA from
SGS), however, a pure spherical shape is also very unlikely to occur in natural snow. To have
a reasonable comparison between satellite-derived SPS and field-measured SPS, the
quantitative information of „roundish" or „irregular" shapes from both satellite and field
measurement communities may be an option. Under this comparison strategy, a „droxtal" shape
derived from satellite observation is somehow identical with a „spherical shape" in field
measurement.
The second and third column in Table 5 are SnowEx17-measured and XBAER-derived
SPS. The abbreviations of the SPS are listed in Table 2. The 4-6[th] columns are the temperature,
wetness of snow and the comments provided by campaign, respectively. Previous publications

show that ice cloud and fresh snow are best described by aggregate of 8 columns (Platnick et al., 2017; Järvinen et al., 2018). Both 9th and 11th February are retrieved to be aggregate of 8 columns because both of them are affected by ice cloud. The first sample on 22nd February is reported to be aggregate of 8 columns and the observation of SnowEx17 is fresh snow. The SPS of the second sample on 22nd February is "facet" while XBAER says "droxtal", indicating possible linkage between XBAER derived "droxtal" and filed measured "facet". It is interesting to compare the SPS for the third sample on 22nd February. The SPSs are round and aggregate of 8 columns for SnowEx17 measurement and XBAER retrieval, respectively. The atmospheric condition is reported to be "windy" and the snow layer is wind-affected and not very well-banded ice crystal. Ice crystal shape in blowing snow is likely to be irregular and aggregated (Lawson et al., 2006; Fang and Pomeroy, 2009; Beck et al., 2018), which is strongly affected by the near surface processes (Beck et al., 2018). Snow grain may also get rounded due to sublimation in blowing snow (Domine, 2009). The wind blowing snow may be well-represented optically by a "aggregate of 8 columns" shape, as retrieved by XBAER.

Table 5 The comparison between SnowEx snow grain shape and XBAER retrieved SGP during February, 2017.

| Date | SnowEx shape | XBAER shape | Temperature (°) | Wetness | Comment |
|------|-------------|-------------|----------------|---------|---------|
| 02-09 | Rounds | col8e | 0.2 | Wet | - |
| 02-11 | New Snow | col8e | -2.5 | Middle | Storm snow, some grapple, some aggregation of crystals |
| 02-22 | New Snow | col8e | -5.1 | Dry | Very surface has sparse surface hoar, affected by yesterday's warm up, bit of crust fragments |
| 02-22 | Facets | droxa | -3.6 | Dry | Very very thin layer of tiny surface facets, still standing not well formed |
| 02-22 | Rounds | col8e | -1.8 | Dry | Surface very wind-affected very thin (3mm) melt- freeze layer not very well-banded |

Table 6 shows the comparison of SSA. For the three cloud-free samples, the difference of
XBAER-derived SSA and SnowEx17 measured SSA is 2.7 m$^2$/kg, which is significantly
smaller than what has been reported by previous publications. For instance, the differences
between satellite retrievals and field measurements are reported to be 9 m$^2$/kg and ~6 m$^2$/kg as
presented in Mary et al (2013) and Xiong et al (2018). An interesting case is observed for the
two-sample on 22$^{nd}$ February. The SGSs show the same values for these two match-ups (both
are 254.4 μm from XBAER and 250 μm from SnowEx), however, ground-based measurement
shows almost two times the difference of SSA (29.8 m$^2$/kg vs 14.6 m$^2$/kg) for these two samples,
which is due to the different SPSs. SnowEx shows that the SPSs are new snow and facets for
these two samples, respectively. XBAER derived SSAs are 24.5 and 12.9 m$^2$/kg, which agrees
well with SnowEx measurement. Since both SnowEx and XBAER provide very similar SGS
(250 μm vs 254.4 μm), the agreement of SSA indicates that XBAER derived "aggregate of 8
columns" is comparable to „new snow" while XBAER derived „droxtal" is somehow
„identical" to "facets" in SnowEx. Cloud contamination introduces an overestimation of SSA,
especially for 11$^{th}$ February. According to the investigation from the companion paper, for
reference SSAs of 37.3 and 25.9 m$^2$/kg, SSA is expected to be ~ 65 m$^2$/kg and >100 m$^2$/kg for
cloud contamination with COT ~ 0.5 and 10, respectively. The real satellite retrieval values are
56.5 and 136.8 m$^2$/kg, respectively.

Table 6 The comparison between SnowEx SSA and XBAER retrieved SSA during February,
2017.

| Date | Lon(°) | Lat(°) | SnowEx(m$^2$/kg) | XBAER(m$^2$/kg) |
|------|--------|--------|------------------|-----------------|
| 02-09 | -108.1092 | 39.0369 | 37.3 | 56.5 |
| 02-11 | -108.0462 | 39.0278 | 25.9 | 136.8 |
| 02-22 | -108.0634 | 39.0444 | 18.5 | 17.4 |
| 02-22 | -108.0625 | 39.0459 | 14.6 | 12.9 |
| 02-22 | -108.0617 | 39.047 | 29.8 | 24.5 |


The above validation for the retrieval of SGS, SPS, and SSA using the XBAER algorithm,
although with limited samples, indicate the consistent of the sensitivity study from the
companion paper in part 1 and the retrieval results in part 2, as presented in this section.

**5.2 Validation using the observations of other campaigns**

For a comprehensive validation, we have analyzed the rest of the sites beside the SnowEx site. The comparison is peformed based on the daily mean observation following the method from Wiebe et al. (2011). We have restircted the SGS in the range of $0 - 300$ μm while the SSA is in the range of $0 - 100$ m²/kg. Thus there may be a slightly difference in the number of total match-up numbers for SGS and SSA. Fig. 8 shows the comparison between XBAER derived snow properties and field-based measurements. Both SGS and SSA show good correlation between XBAER derived and field-based measurements, with correlation coeffcients larger than 0.85. A clear underestimation of SGS, especially for large SGS values, is observed. This can also been seen from the slope of the regression (slope = 0.67). XBAER shows good agreement with field-based measurements, especially for SGS smaller than 150μm. The underestimation occurs mainly over regions with complicated surface condition and/or large aerosol loading. In general, we can see larger deviation to the 1:1 line when AOT values are larger. This agrees with a major finding in Part 1 of the companion paper, that is aerosol contaminaton introduces underestimation of SGS. For instance, large AOT values can be seen over China, while strong underestimation of SGS is also observed. For Alps and two canadian (Canada-Alex, Canada-Josh) sites, the AOT values are farily low, the underestimation may be explained by the strong surface inhomogeneity (possibly due to different surface types in one satelite pixel). For site Greenland and Antarctica, where AOT values are low and surface is covered mainly by snow, XBEAR shows good performance. This can be confirmed by the RMSE values. The RMSE values in Fig. 8 are calculated only for site Greenland and Antarctica, to avoid the large outliners over other sites (please be noted other sites provide quite limited number of match-ups, see Fig. 9). The RMSE value is 12 μm.

The comparsion between XBAER derived and field-measured SSA shows no significant under/over-estimation (slope = 1) with correlation coeffcient R = 0.93. XBAER derived SSAs are, in general, larger than field-based measurements. This can be explained by the use of different SPS assumptions. In the XBAER algorithm, for the match-ups shown in Fig. 8, most SPSs are non-convex while the convex SPS is used for field-measured values. We recall, that for the same SGS, non-convex particle leads to a larger SSA, compared to convex particle. The

impact of aerosol contamination, compared to surface condition, seems to play a major role of
the observed overestimations.
The potential linkage beween XBAER derived SPS and filed-measured SPS is also
presented in Fig. 8. This is named as SPS similarity in this manuscript. The SPS similarity is
defined as the ratio of match-up number for a given SPS pair (XBAER retrieved Yang SGS,
field measured ICSSG SPS) to the total match-up number. The higher SPS similarity, the higher
chance this SPS pair may occur in reality, indicating the higher possibility of the retrieved Yang
SPS may have closer relationship with ICSSG SPS. According to Fig. 8, we can see that
aggregate of 8 columns, solid bullet rosettes and column show stronger linkage with the
rounded grains while droxtal, plate and column show stronger linkage with the faceted crystals.
This may lead to some imperfect and highly uncertain linkage between XBAER derived SPS
and the ICSSG SPS. Aggregate SPS in XBAER is likely to be matched with rounded gains
while single SPS in XBAER is possibly linked to faceted crystals. There are also possible
linkage between XBEAR SGS and ICSSG SPS, for instance, aggregate of 8 columns and plate
with precipitation particles, solid bullet rosettes with depth Hoar, droxtal and plate with surface
hoar. The above linkage also indicates that aggregate of 8 columns (linked to rounded grains
and precipitation particles) may represent fresh snow while droxtal (linked to faceted crystals
and surface hoar) may represent aged snow. This agrees with the previous analysis over
Greenland.

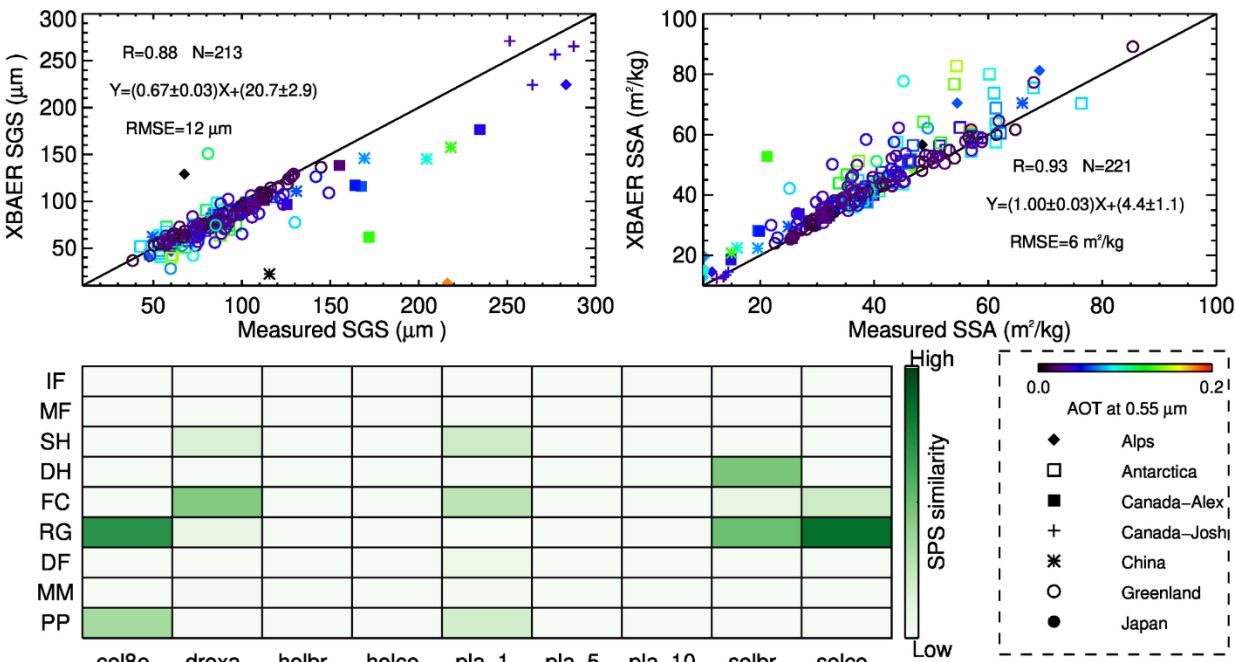


Fig 8 Validation of XBAER derived SGS, SPS and SSA. The upper panel shows the scattering
plot for SGS and SSA, while the lower panel shows the relationship of SPS between XBAER
and ICSSG. The match-ups for SGS and SSA are distinguished by sites and the AOT. The
correlation coefficient (R), number of match-ups (N), the regression equation, and the RMSE
are given. The relationship of SGS between XBAER and ICSSG (named as SPS similarity) is
defined as the ratio of the number given match-ups to the total match-ups.

Fig. 9 and Fig. 10 show the time series of SGS and SSA over each site. We can see that
sites Greenland and Antarctica provide most of the match-ups. Both SGS and SSA show good
agreement between XBAER derived and field measured values over these two sites. For SGS,
the correlation coefficients are 0.85 and 0.89, the RMSEs are 14 and 9 $\mu$m, respectively. For
SSA, those values are 0.84, 0.89 for correlation coefficient and 8 and 7 $m^2$/kg for RMSE,
respectively. Although the other sites provide limited match-ups, they still give helpful
information for the understanding of impacts of surface and atmospheric conditions. In general,
sites China and Japan show large AOT values, leading to underestimation of SGS and
overestimation of SSA. For two Canadian sites (Canada-Alex, Canada-Josh), the under/over-
estimation of SSA and SGS may largely explained by the surface condition. Site Alps seems to
be affected by both surface and atmospheric impacts.

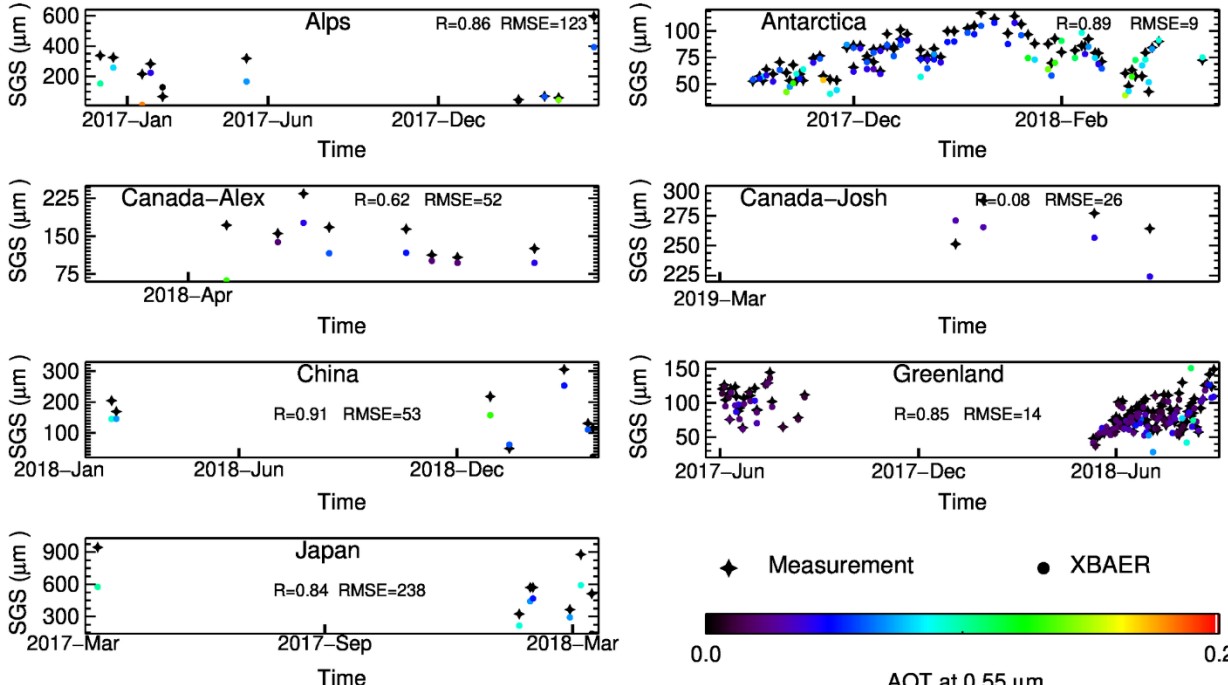


Fig 9 Time series of XBAER derived and field-measured SGS for each site. The match-ups for
SGS are distinguished by the AOT values. The correlation coefficient (R) and the RMSE are
given.

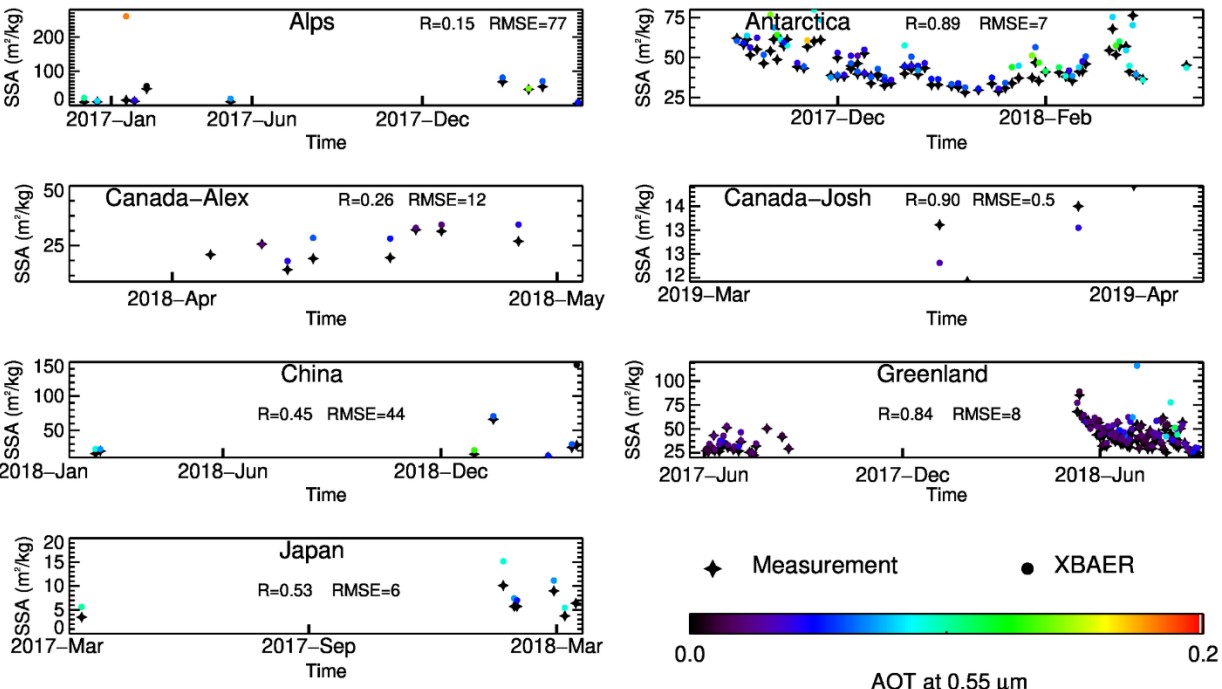


Fig 10 Time series of XBAER derived and field-measured SSA for each site. The match-ups
for SGS are distinguished by the AOT values. The correlation coefficient (R) and the RMSE
are given.

## 5.3 Validation using the observations of aircraft campaign

The optical snow grain size over Arctic sea ice was derived from airborne SMART
measurements as described in Sect. 2.3. Fig. 11 (a) shows the retrieved grain size along the
flight track (black encircled area) taken on 26 March 2018 between 12 and 14 UTC north of
Greenland. During this period of cloudless conditions, a Sentinel3 overpass (12:29 UTC)
delivered SGS data based on the XBAER algorithm as displayed in the background of this map
with 1 km spatial resolution. In general, lower SGS were observed by both methods in the
vicinity of Greenland, while in particular in the North-East region of the map (red dashed circle
in Fig. 11 (a)) SGS values of up to 350 μm were derived from the aircraft albedo measurements.
Also the XBAER algorithm reveals higher values in this region. For a direct comparison
XBAER data were allocated to the time series of the SMART measurements along the flight
track. Afterwards all successive SMART data points assigned to the same XBAER location
were averaged to compile a joint time series of both data sets as displayed in Fig. 11 (b). Overall
a correlation coefficient of R = 0.82 and a root mean squared error of RMSE = 12.4 μm was
derived, where SMART (mean SGS: 165±40 μm) generally shows higher grain sizes than
XBAER (mean SGS: 138±21 μm). The course of the SGS follows a similar pattern for both
methods, with largest deviations when the aircraft measured in the red dashed circled area from
Fig. 11 (a). The corresponding time periods are indicated by the light red shaded area. Camera
observations along the flight track have revealed an increase of surface roughness in this area.
Note, that the flight altitude varied for the flight section shown in Fig. 11 (a). Due to the low
sun, such a non-smooth surface produces a significant fraction of shadows which lowers the
measured albedo. Consequently, the retrieved SGS is affected in particular for the lowest flight
section when SMART collects the reflected radiation with high spatial resolution. This might
explain why the deviation of the retrieved SGS values in this area are largest around 13 UTC
when flight altitude was in the range of 100 m.

The SGS retrieval based on the algorithm suggested by Zege et al. (2011) and Carlsen et

al. (2017) give the optical radius of the snow grains, such that the SSA can be derived applying
Eq. (A1) from companion paper. The map of the SSA (Fig. 11 (c)) reflects a similar pattern
than observed for the SGS, showing an inverse behavior to Fig. 11 (a). In average, XBAER
(mean SSA: 24±3 $m^2$/kg) and SMART (mean SSA: 21±5 $m^2$/kg) agree within the 1-sigma
standard deviation. The correlation of SSA between XBAER and SMART is similar as for the
SGS with a correlation coefficient R = 0.81 and RMSE = 2.0 $m^2$/kg. A comprehensive
comparison between XBAER and SMART is given in Jake et al. (2021).

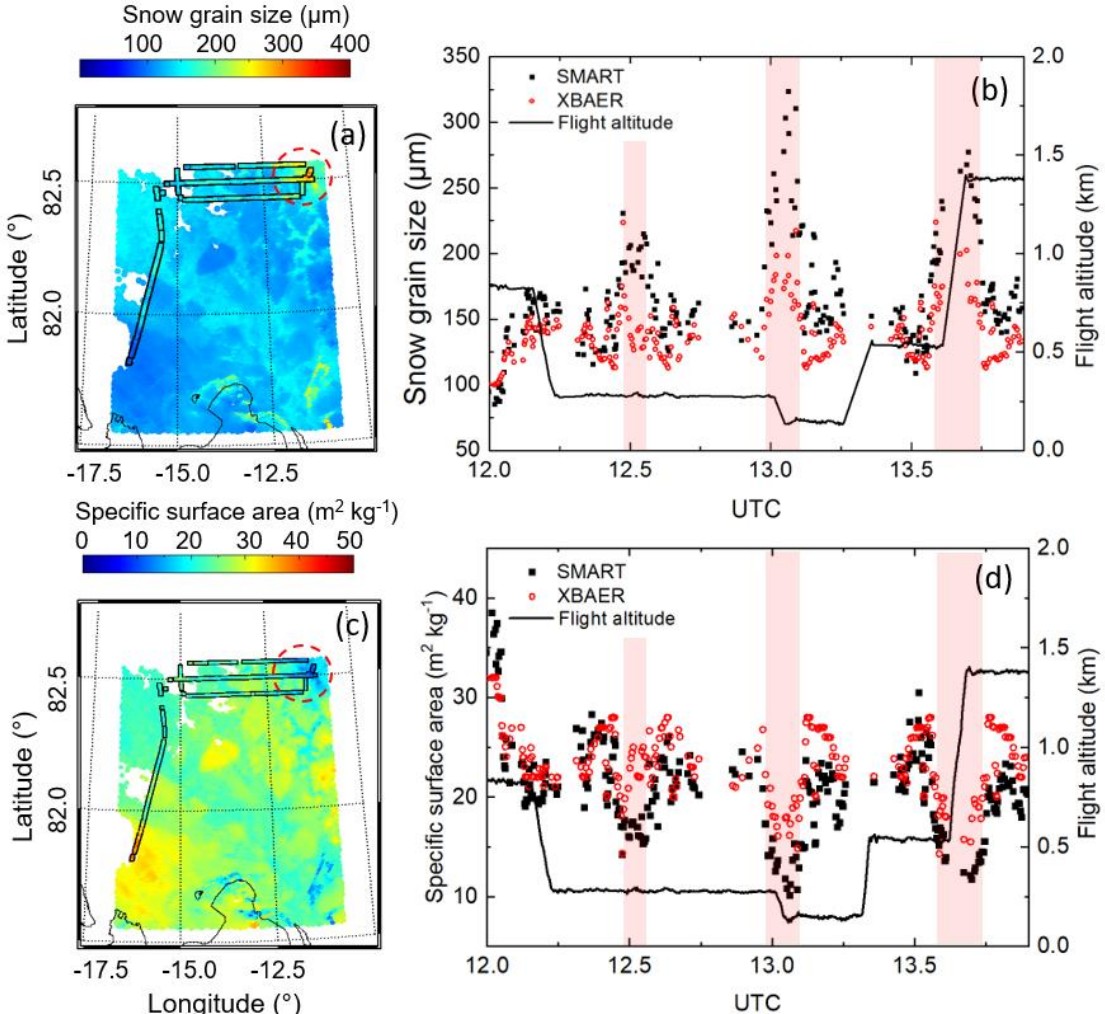

Figure 11: (a) Map of SGS retrieval results from Sentinel measurements in the North of Greenland from 26 March 2018. The black encircled area represent the SMART retrievals of the SGS along the flight track. The red dashed circle marks a region with increased surface roughness. (b) Time series of both retrieval data sets adapted to the aircraft flight path. Periods matching with the circled area in (a) are shaded in light red. (c) and (d) are similar to (a) and (b) but for SSA. Additionally, the flight altitude is given.

Since XBAER is also designed to support MOSAiC campaign on an Arctic-wide scale (Mei et al., 2020c), it is important to have an overview of how snow properties look like on an Arctic-wide scale for existing campaign. Fig. 12 shows the SGS, SPS and SSA geographic distribution over the whole Arctic for 26 March 2018. Northern Greenland, North America, and central Russia show large snow particles, especially over North America. And the SPS shows

more diversities in lower latitude compared to the central Arctic, indicating stronger SPMP. An
aggregated shape such as aggregate of 8 columns is the dominant shape in the central Arctic
while column is one of the dominant shapes in lower latitude. SSA shows large values in the
lower latitude Arctic (northern Canada, southern Greenland, western Norway, southern Finland,
northern Russia) while the values are smaller in the central Arctic.

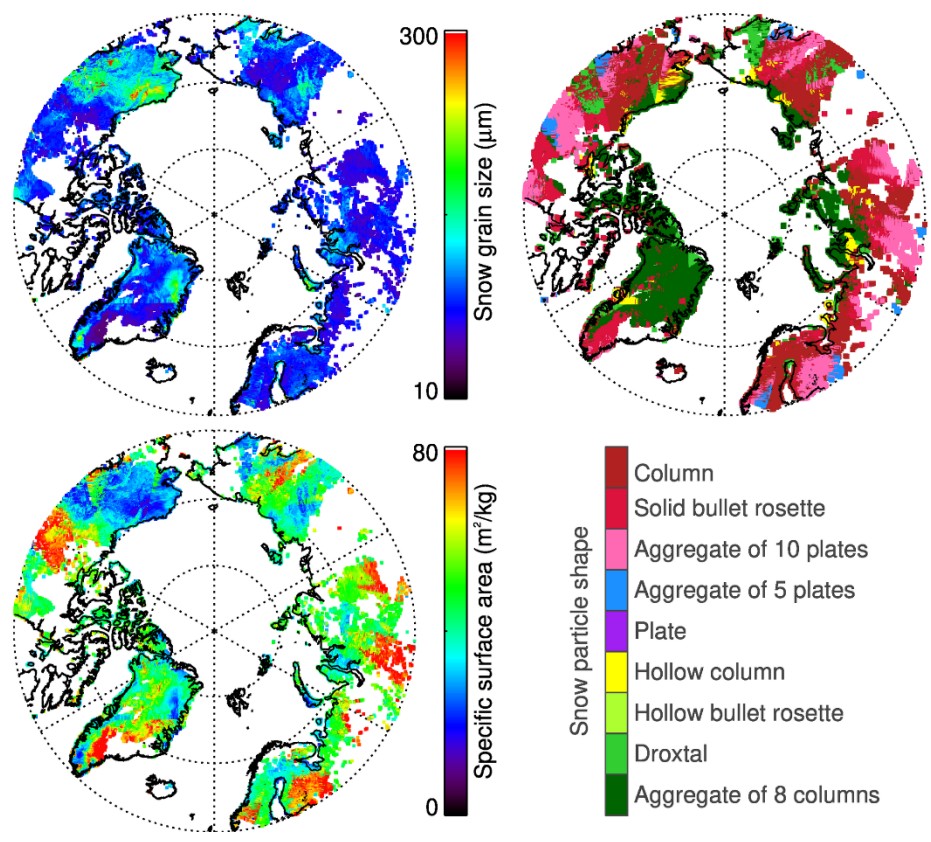

Fig. 12 The distribution of XBAER-derived SGS, SPS and SSA over the whole Arctic for 26
March 2018

**6 Discussion**
The above analysis shows the promising quality of XBAER-derived SGS, SPS and SSA results.
The XBAER retrieved SGS, SPS and SSA can be used to understand the change of snow
properties temporally. Even though the snow metamorphism depends on the environmental
conditions, Aoki et al. (2000) and Saito et al. (2019) pointed out that a 4-days time scale is a
reasonable time-span to see the temporal change of snow properties. Fig. 13 shows XBAER-
derived SGS (upper panel), SPS (middle panel) and SSA (lower panel) over Greenland during
27 – 30 July, 2017. Large variability for SGS, SPS and SSA can be seen during these four days,
indicating the impacts of snow metamorphism on the snow properties. Fig. 13 shows snow
melting process in both western and northeastern parts of Greenland, especially during 28 July.
The strong melting in July over Greenland has also been reported by Lyapustin et al (2009).
SPS over southeastern part of Greenland becomes smaller during those four days. No snowfall
has been reported according to POLAR PORTAL report
(http://polarportal.dk/en/greenland/surface-conditions/) during these four days, thus the smaller
SGS may be caused by local snow metamorphism process and/or due to the wind-blown fresh
snow, transported from central Greenland to southeastern parts. This is consistent with the wind
direction as presented in Fig. 14. The wind speed is over 6 m/s, which is strong enough to blow
the surface ice crystal up. However, possible cloud containmination over northwest of
Greenland may occur, leading to very small SGS. The change of SGS is also consistent with
the change of SPS. Please be noted, since the SGS and SPS are retrieved simultaneously, the
selection of different SPSs leads to a different SGS, thus the change of SGS and SPS with
respect to time may also be affected by the algoritm itself. According to Fig. 13, SPSs over
Greenland derived from the XBAER algorithm are mainly droxtals and solid bullet rosettes for
the selected days. The solid bullet rosettes and droxtal are typical ice crystal shapes for fresh
snow and aged snow (Nakamura et al.,2001), respectively. The wind-blown fresh snow might
be transported to the eastern part of Greenland, and fresh snow covers the original aged snow,
thus a solid bullet rosettes shape is retrieved. According to Fig. 8, droxtals and solid bullet
rosettes retrieved by XBAER may link to faceted crystals and rounded grains in ICSSG,
respectively. During the transport, faceted crystals turn into rounded grains. The change of SSA
follows the change of SGS and SPS. SSA over central Greenland is larger while it is smaller in
the coastline regions. This can be explained by the reduced SPMP impact on the snow
properties due to the increase of elevation in central Greenland. Inversely proportional to SGS,
the SSA reduces. The coverage of large SSA over the eastern part of Greenland increase during
these four days, indicating the "snowfall" feature due to transport. This wind-induced transport
feature, similar to fresh snowfall, changes both SGS and SPS. And this process is revealed by
and superimposed on the SPMP during the temporal change of SSA retrieved from satellite
observations (Carlsen et al., 2017).

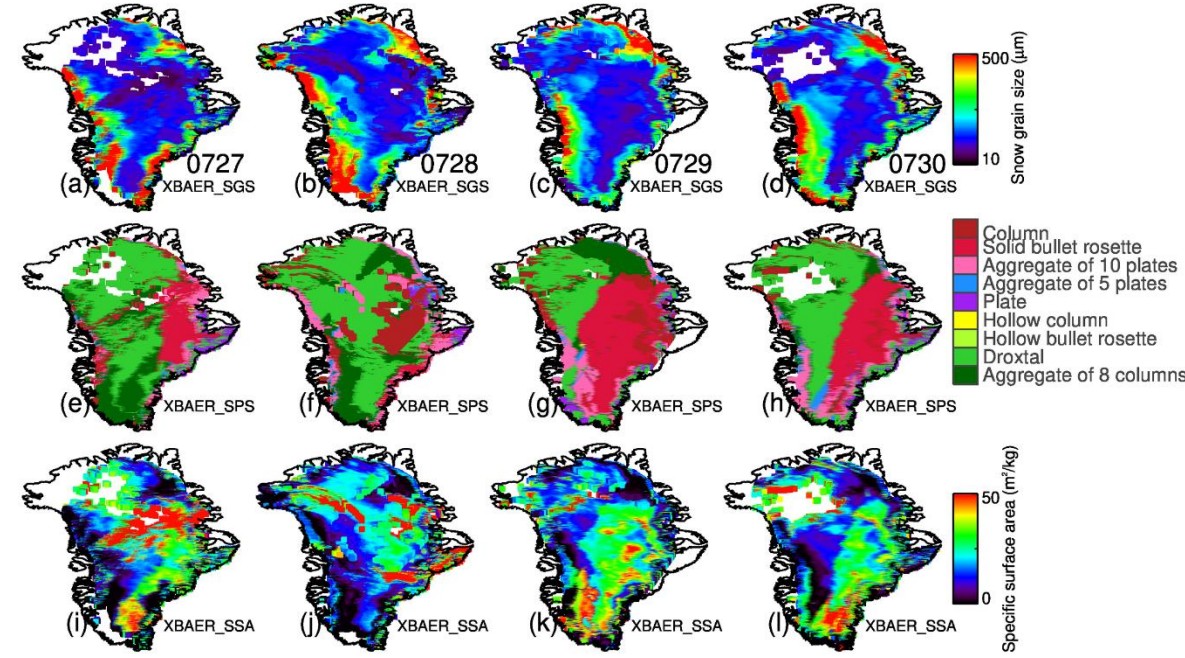


Fig 13. XBAER derived SGS, SPS and SSA over Greenland during 27 – 30 July 2017.

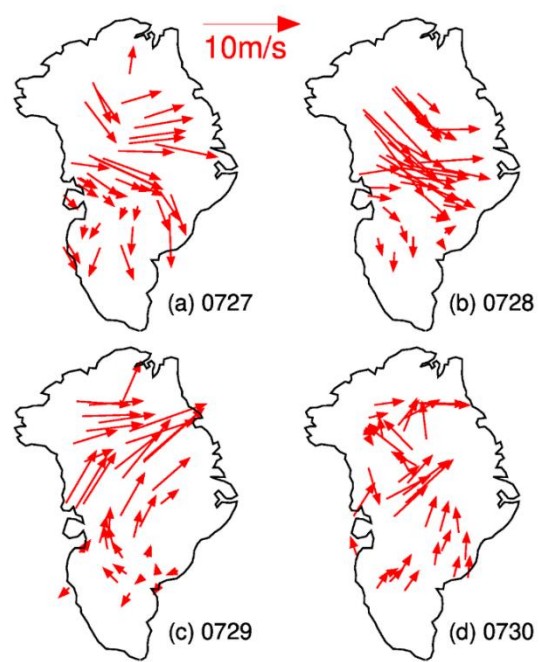


Fig 14. Wind direction (reference to North) and wind speed (unit: m/s) over Greenland during
27 – 30 July 2017.

## 7 Conclusions

SGS, SPS and SSA are three important parameters to describe snow properties. Both SGS, SPS and SSA play important roles in the changes of snow albedo/reflectance, further impact the atmospheric and energy-exchange processes. A better knowledge of SGS, SPS and SSA can provide more accurate information to describe the impact of snow on Arctic amplification processes. The information about SGS, SPS and SSA may also explore new applications to understand the atmospheric conditions (e.g. aerosol loading). Although some previous attempts (e.g. Lyapustin et al., 2009) show the capabilities of using passive remote sensing to derive SGS over a large scale, no publications have been found to derive SGS, SPS and SSA simultaneously. This is the first paper, to our best knowledge, attempting to retrieve both SGS, SPS and SSA using passive remote sensing observations.

The new algorithm is designed within the framework of XBAER algorithm. The XBAER algorithm has been applied to derive SGS, SPS and SSA using the newly launched SLSTR instrument onboard Sentinel-3 satellite. The cloud screening is performed with a synergistical technique using both OLCI and SLSTR measurements. The synergistical cloud screening in XBAER is easy-implementable and effective-runable on a global scale, with high-quality, enables a cloud-contamination-minimized SGS, SPS and SSA retrieval using passive remote sensing.

Besides the cloud screening, another pre-process is the atmospheric correction. Aerosol plays a non-ignorable impact on the retrieval of SGS, SPS and SSA, even over the Arctic regions, where aerosol loading is small (AOT, at 0.55μm is around 0.05) (Mei et al., 2020b). In the XBAER algorithm, the MERRA simulated AOT at 0.55μm, together with a weakly absorption aerosol type (Mei et al., 2020b) is used as the inputs for the atmospheric corrections.

The SGS, SPS and SSA retrieval algorithm is based on the publication by Yang et al (2013), in which a database of optical properties for nine typical ice crystal shapes are provided. Previous publications show that this database can be used to retrieve ice crystal properties in both ice cloud and snow layer (e.g.,Järvinen et al., 2018; Saito et al., 2019). The algorithm is a LUT-based approach, in which the minimization is achieved by the comparison between atmospheric corrected TOA reflectance at 0.55 and 1.6 μm observed by SLSTR and pre-

calculated LUT under different geometries and snow properties. The retrieval is relatively time-
consuming because the minimization has to be performed for each ice crystal shape and the
optimal SGS and SPS are selected after the 9 minimizations are done. The SSA is then
calculated using the retrieved SGS and SPS based on another pre-calculated LUT.
The comparison between XBAER derived SGS, SPS and SSA show good agreement with
the SnowEx17 campaign measurements. The average absolute and relative difference between
XBAER derived SGS and SnowEx17 measured SGS is about 10 μm and 4%, respectively.
XBAER derived SGS also shows good agreement with MODIS SGS product. XBAER
retrieved SPS reveals reasonable and explainable linkage with SnowEx17 measurements. The
difference of XBAER-derived SSA and SnowEx17 measured SSA is 2.7 $m^2$/kg. The retrieval
results over Greenland reveal the general patterns of snow properties over Greenland, which is
consistent with previous publications (Lyapustin et al. 2009). The change of SGS, SPS and SSA
on a 4 days time span is also observed using XBAER retrieved SGS, SPS and SSA. The
comparison with aircraft measurement during PAMARCMiP campaign held in March 2018
also indicates good agreement (R = 0.82 and R=0.81 for SGS and SSA, respectively), XBAER-
derived SGS and SSA reveal the variabilities of the aircraft track of the PAMARCMiP
campaign. A intensive validation is performed using seven addtional field-based measurements.
XBAER derived SGS and SSA show high correlation with field measurements, with correlation
coeffcients are higher than 0.85. The RMSE for SGS and SSA are less than 15 μm and 10 $m^2$/kg,
respectively. The validation of SPS reveals that XBEAR derived aggregate SPS is likely to be
matched with rounded grains while a single SPS in XBAER is possibly linked to faceted
crystals in the ICSSG classification. This possible linkage, although inaccurate, will be helpful
to understand the snow properties in a large scale.
Although the presented version of the XBAER retrieval algorithm shows promising results,
we see at least four possibilities to improve its accuracy. Potential cloud contamination may
still occur according to the analysis, exploiting the time-series technique, as described in
Jafariserajehlou et al. (2019). Currently only single ice crystal shape is used in the retrieval, the
mixture of different ice crystal shapes i.e., the snow grain habit mixture model (e.g., Saito et al.
2019) will be tested in further work. Another potential improvement may be linked to the use
of polydisperse ice crystals (e.g. gamma distribution). The potential impacts of the vertical
structure of SGS and SPS also need to be investigated in the future.
XBAER-derived SGS, SPS, and SSA will be used to support the analysis of MOSAiC
expedition and other campaign-based measurements (Jake et al., 2021).

**Code and data availability**
The data over Antarctica site is provided by Dr. Ghislain Picard. The data over site Greenland
site is provided by Dr. Hans Christian Steen-Larsen. The data over site Canada-Alex site is
provided by Dr. Alexandre Langlois. The data over Canada-Josh site is provided by Dr. Joshua
King. The data over China site is provided by Dr. Tao Che. The data over Japan site is available
at https://doi.pangaea.de/10.1594/PANGAEA.909880. The data over Alps site is available at
https://perscido.univ-grenoble-alpes.fr/datasets/DS330. The data over SnowEx site is available
at https://nsidc.org.
**Author contributions**
LM and VR conceptualized the study, LM implemented the code and processed the data. LM,
VR EJ, XC analyzed the data. LM prepared he manuscript with contribution from all co-authors.
LM, VR, MV and JB polished the whole manuscript.

**Competing interests**
The authors declare that they have no conflict of interest.

**Acknowledgements**
This research was funded by the Deutsche Forschungsgemeinschaft (DFG, German Research
Foundation) – Project-ID 268020496 – TRR 172. The comments by Dr. Alexandre Langlois,

Dr. Ghislain Picard and the anonymous reviewer help to improve the quality of the manuscript significantly. The authors highly appreciate the effort from Dr. Adam Povey (University of Oxford) to help to deal with the huge amount of the SLSTR L1 data. The authors would like to thank Prof. Knut von Salzen from Environment Canada for the valuable discussion. We thank the support from Dr. Lisa Booker from National Snow and Ice Data Center, Boulder to understand the SnowEx17 campaign data. We thank Dr. Alexander Kokhanovsky from VITROCISET, Darmstadt, Germany and Prof. Jason E. Box from Geologic Survey of Denmark and Greenland (GEUS) for the valuable discussion. We thank Salguero Jaime for providing the MODSCAG snow products. The MODIS snow product data are provided by MODSCAG team and SLSTR/OLCI data are provided by ESA.

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
