# Peer review of "The retreival of snow properties from Sentinel-3 SLSTR - part 2: results and validation"

_The Cryosphere, 2020_

## Referee Comment (RC1) · Ghislain Picard (Referee) · 10 Nov 2020

Review "The retrieval of snow properties from SLSTR/Sentinel-3 -part 2: results and validation" by Mei and colleagues

The paper aims at validating an algorithm to retrieve snow grain size and shape, and snow specific surface area from the space-borne SLSTR sensor. The algorithm was described in another paper in review (companion part 1), the present manuscript is dedicated to the validation.

The overall goal of these two parts is of interest for the cryosphere community, in particular because SLSTR is on the Sentinel 3 series of satellite which will be able for decades. The paper is original and clear. Nevertheless, my recommendation is to

postpone the acceptance of this paper for three main reasons:

- The validation is based on a too limited set of in-situ data, part of it is discarded because of cloud contamination (SnowEX17). The text truly dedicated to the algorithm performance evaluation is also relatively short and seems unfinished, most of text is about the difficulty to perform the validation, which in the end does not contribute to give confidence in the retrieval algorithm. The conclusion about the algorithm performance therefore lacks of support. There are also several technical issues (see below in the detail comments) in particular one on the RMSE definition. The lack of datasets is a common problem, but not to the extent depicted by the authors. The main example is for the snow grain size. The manuscript cites Kokhanovsky et al. 2019 which pursues a very similar objective as the present manuscript but uses OLCI, (on Sentinel 3 as well as SLSTR) to estimate grain size and SSA (not the grain shape). For the validation these authors used an extensive dataset with 100s of SSA field measurements in Greenland and in Antarctica. These data can be either retrieved from the graphs or in principle obtained from the authors, and should be used here to complete the validation (or even replace the 3 SnowEx measurements). Moreover the performance between the SLSTR and OLCI algorithms could be analyzed at these in-situ points. At last, the authors "emphasize that the results presented in this section is considered as preliminary" (L373). They indeed propose to include Mosaic data in their analysis in the future. My concern is whether it is worthwhile for the community to publish "preliminary results" in two papers. My suggestion is indeed to wait for complete results and include Mosaic dataset.

- the grain shape is a big issue of this study. It is claimed to be a major advantage compared to other algorithms (e.g. L617) but the demonstration is missing. First because it is difficult if not impossible to validate. I acknowledge that snow shape is a difficult topic. However as for the validation of the grain size, the choices of the authors are limiting the ability to perform the validation. The algorithm assumes and retrieves geometrical shapes that are representative of precipitating crystals, not of snow on the

ground although the algorithm is supposed to be used for snow on the ground. A first consequence is that the algorithm can not perform well, because the phase function of such shapes does not apply to snow on the ground (expect for fresh snow). Snow on the ground is usually more rounded and irregular than crystals in the atmosphere. The second consequence (and the main one) is the difficulty to perform the validation. Data recorded by snow practitioners and scientists in the field usually follows the international classification of seasonal snow on the ground (Fierz et al. 2009, not cited in the manuscript) which has some shortcomings but is widely used. Since the algorithm does not use these "standard" shapes, it is inherently impossible to perform a fair comparison with external data. It follows a third consequence about the usefulness of the shape information retrieved by the algorithm. I'm wondering how useful is this retrieved "grain shape" for snow community since it does match with its standards. I suggest that to solve this major issue, ideally by adapting the shapes used by the algorithm, and if not possible at least by establishing a link between the different shape systems. Even if imperfect and highly uncertain, this link will benefit to the whole clarity of the paper and will help to shorten the validation section (see comments below). They should also explain why retrieving the shape is useful for the algorithm. The algorithm uses a first guess grain size from another algorithm but no comparison is given. I would expect the authors to demonstrate that taking into account the grain shape has an effective positive impact on the SSA or grain size estimates. This would be very useful for the snow remote sensing community to know if such an approach is fruitful.

- the benefits to split the study in two parts is not clear. The paper (part 2) presents the validation of an algorithm that is not described, which raise several questions and make it be difficult to read without reading the other paper (part 1). For the review, I didn't read the part 1 (I just browsed it) to be in the same position as a normal reader. I found that reading part 2 was difficult with many open questions about the algorithm and was sometimes annoying because of a few elusive statements referring to the part 1 without providing information. E.g. "The similarities and differences of the required snow parameters and their accuracy between the snow remote sensing community

and other communities (e.g. field-measurement community) are detailed discussed in part 1 of the companion paper (Mei et al., 2020), thus we will not summery again in this paper. ". The length of this part 2 is normal and the information density is relatively low. For the comfort of the reader, I suggest to shorten or remove some sections (e.g. the first results section on Greenland), and merge with the part 1. Only if extending the validation as proposed above with a complete dataset and with Mosaic data, it would be justified to make two papers.

Detailed comments:

L63. What is the definition of "grain size" used here ?

L 69: correct "detailed discussed"

L70: "summery" → "summary"

L91-L92: I'm not sure to understand "to be with good quality"

L98-L99. Please add a reference / name for the operational product.

L104 I'm not sure to understand "to partly taking snow irregular ".

L118: "Details of these issues have been discussed in Part 1 of the companion paper.". Please remove and add a proper reference. Or just remove.

L120-122: This sentence is strange, "no publication. . . especially using" seems contradictory.

L 124-126. I don't understand the sentence. What is an "optimal complex shape". The part 1 paper seems to use very geometrical/simple shapes and the goal of the retrieval algorithm is to retrieve SPS. How does this apply to this sentence ? Also, what do you mean by the e.g. TOA ?

L147-149. I suggest to move this statements to the conclusion.

L150. I suggest to remove this statement or merge the two papers.

L152 – L162. I suggest to move this paragraph to the discussion because it is a typical analysis of the uncertainties of the results/validation. The representativeness issue is a general problem, that affects any in-situ vs remote sensing comparison. Why the SPS would be particular ? This also concerns SGS and SSA.

L170. Remove double "show".

L215. I suggest to remove SnowEx17 grain shape from the table because it is misleading even with the warning in the legend caption. Instead it is possible to list these grain type in the caption and/or in the main text. Note that the grain type measured by SnowEx17 are not specific to this campaign but refer to the international classification (Fierz et al. 2009).

L253 have→are

L276. Could you give a definition of spherical albedo and Lambertian surface albedo ?

L281. Could you indicate the resolution of MERRA ?

L282. Our → a

L339-355. The comparison is very qualitative and referring to generic and broad "classification" of "polar snow" does not bring significant information for this validation, especially because not all the existing references about snow grain shape and size have been taken into account. It must be taken into account that July is warm with a large proportion of the ice-sheet subject to melt, which unequivocally leads to rounded coarse grains very quickly.

Because the validation can not be done with information that are not available, I suggest to convince the reader that the results are plausible using cross-analyzed external data: - use MERRA to separate where the snow is fresh and for which the present discussion in these lines apply fairly well. - where snow is fresh use successive image to show that SGS increases (and SSA decreases) as predicted by metamorphism (as you suggest, July is interesting for the most rapid metamorphism). - use passive

microwave (or MERRA or SLSTR thermal channels) to separate where melt is active and where the grains are very likely to be rounded. - use the images next 28 July 2017 to demonstrate that the blue shape for instance in NW Greenland are not due to clouds/aerosols. (I've made this comment before reading the discussion, see further comments below).

I also suggest to mask out areas in the ablation zone with ice and dirty snow, as the algorithm does not work in these cases. This should be emphasized.

Fig 3. adding a scatterplot with relevant statistics (R2, RMSE, bias, . . .) is common for a more quantitative validation. In particular, it would be useful to compute the same statistics with the first guess to really show the benefit of the algorithm.

L371. The previous section was titled "Results" but was also a comparison (and validation to some extent). Why not a unique Result section that includes both comparison ?

L372. I suggest to remove "validate". L373. ground-based/aircraft → ground-based and aircraft

L377- 379. I'd remove this introductory sentence that starts by concluding that the algorithm is good although the actual goal of the present sessions is to perform the validation.

L 385. "time and location" or "times and locations".

L394. Why the rows are not sorted chronologically as in next figure ? What is the order ? Has the gray shade in the last row a meaning ?

L398. This is the second "Fig 4". Review numbering. +Please add a scale to the maps.

L406. How does this perform in the case of thin clouds ?

Fig 4 and Fig 5. I don't understand why two figures ? If I understand well, Fig 4 is a zoom of Fig 5 ? They should be merged in a single composition using the same

symbology / graphic style.

L412-413. "is not correctly avoided". This is a bit confusing. The next sentence is clearer to me but seems to be in contradiction with Table 3 indicating "cloud contaminated snow" for this date (which seems accurate based on Fig 4). L388 indicates that the comment in Table 3 is obtained with the algorithm. Please clarify.

L413. Give → gives.

L421. Add a ref to the study.

L442. "Our → a" or "our calculations with"

L444. Fig 1 → fig 1 with a lowercase as it is referring to another paper. Add the ref.

L451-452. "cloud effective radius" → "cloud ice crystal effective radius". SGS and "ice crystal size" are used interchangeably in the paper which is sometimes (and especially here) confusing.

L464. ""This is similar to the issue in field measurements." what do you mean ? L465. " (e.g.,the measurement of SSA),". This is generally not true. Do you refer to a precise device and processing ?

L466-470. I'd suggest to define in the method section (Table 2) the most-likely correspondence between Yang's shapes and the snow type defined in the international classification (that used in SnowEx) so it is possible here and in the Section 4 in the results section to assess the algorithm performance in a more rigorous way.

L473. "A previous publication" or cite more than one L474 are → is

L479 "is 'facet' while XBAER says 'droxtal' both tend to be roundish". Facets according to Fierz et al. 2009 is not rounded. If the retrieval algorithm SPS can not distinguish rounded grains from faceted grains because both are droxtal, how useful it this for field practitioners ? This asks an important question that is not addressed in the introduction: why and for what usage to retrieve SPS from satellite ?

[Figure]

L483. I do not agree. It is also believed that grains get rounded due to sublimation in blowing snow (Domine, 2009). This probably depends on the conditions, on the actual grains available on the surface, and the strength and duration of the saltation/reptation process.

L493-496. Please indicate the number of points of each comparison (n=...) and the statistical significance of the results. By "difference" do you mean "rms difference" or "difference of the average" ?

L548. Here it would be particularly interesting to see how good the first guess predictor of SGS. I'm really interested by knowing if the algorithm sophistication is worthwhile.

L533. I'm not sure to understand how the RMSE is calculated. The RMSE includes both systematic and random errors, and here given the difference of the mean, the RMSE should be at least 165 – 138 = 17 microns while the text indicate 12 microns. Please check also "lower grain sizes".

L550. The same question applies for SSA, with a difference in the mean of 3 m2/kg, it is not possible that the RMSE is 2 m2/kg.

Fig 8. This figure is interesting but should be used earlier in the validation to infer the errors of estimations. I see the following possible artifacts: - The presence of undetected clouds in the NW Greenland. - The dramatic grain size decrease after 28 July in Eastern Greenland (analysis around L588) is very suspicious and stronger evidences are needed to prove that it would be related to a massive drift event, and not to a retrieval artifact. In particular it would be necessary to demonstrate that the wind sustained over 6m/s for a sufficient long period of time to really bring sufficient quantities of small grains over the considerable distance. - Why grain shape changes so fast between a Droxtal to a column in central Greenland ? Wind is able to drift fresh snow, but in the absence of recent snowfall, if snow was already Droxtal at the surface, wind can not transform it into more elongated crystals. Faceting of grains at such a pace is suspicious. - The Western side is also affected by the grain size change. The

shape change is also marked and different from that observed in the Eastern side. Why this is not discussed ?

L590. The weblink does not point to any data. A figure should be added in the supplementary with direction and wind speed.

---

## Referee Comment (RC2) · Anonymous Referee #2 · 24 Dec 2020

General Comments

This paper describes the results and validation of the XBAER algorithm that retrieves snow grain size (SGS), specific surface area (SSA), and particle shape (SPS) from Sentinel-3 SLSTR instrument. The paper presents the results and evaluate them using the MODSCAG product, in-situ measurements from SnowEx17, and airborne-based retrievals. The validation for cloud-free and partial cloud cover shows promising results from the XBAER algorithm; however, there are some issues related to the validation process and the paper's writing structure.

Regarding the validation process, the main negative point is that the authors state that these are preliminary results and that they are waiting for more data from the MOSAiC project to increase the number of observations for validation. However, this paper

accompanies another paper on the development of the XBAER algorithm. If these are preliminary results, it would make more sense to fit some of these results in the first paper, and then wait for the MOSAiC data to submit a more comprehensive validation in the second paper. I understand that it is hard to obtain enough data for remote sensing validation, but since there is ongoing data collection, the wisest decision will be to wait until the MOSAiC dataset is fully collected.

Regarding the paper's writing structure, in general, the paper lacks conciseness. A few general comments about this topic are the following: - There are long sentences on multiple occasions. - The authors should be more careful about using quantitative adjectives when describing their results or other authors' results. I would recommend using the actual number instead. - I would suggest that the authors make a thorough revision of the use of articles, prepositions, and verb agreements in the paper. - The purpose of this paper should be more clearly stated in the introduction. - There is excessive use of quotes. - The discussion paragraph is too long and speculative. There is a lot of discussion in the results section already. I would recommend the authors to create a section for results and discussion together instead. - The conclusion is too long and with too much redundancy. It should state the main findings, limitations, and future studies, and if it was able to meet the goals of the study. In addition, the main findings in the conclusion should follow the same order that the results are presented. - More detail is provided in the Specific Comments and Technical Corrections sections below.

Other general comments on the scientific soundness of the paper are the following: - How have you dealt with the forest in the Grand Mesa site? - One way to improve the SGS validation would be perhaps to extend the validation using more MODSCAG scenes. - It is not appropriate to use full cloud cover field measurements to validate a remote sensing retrieval that only works with cloud-free conditions. Using partially covered skies might still be a reasonable assumption, as long as the limitations of the retrievals under these conditions are addressed, but not full cloud cover as on Feb. 11,

2017. Retrieving snow properties for full cloud cover only shows that your model is able to characterize the properties of cloud ice crystals, but that has no implications for snow properties on the ground.

Specific Comments

Line 14: The OLCI instrument was not used directly to retrieve snow properties; therefore, it would be better to only mention it when talking about the cloud screening process. Line 52: Melting snow does have lower albedo, but the main mechanism that decreases albedo is actually the absence of snow cover. Line 62: The terms field-based and in-situ are synonyms; there is no need to use them together. Lines 62 to 67: Consider splitting or rearranging this sentence to improve its clarity. Line 82: When you mentioned snow fraction. Did you mean snow cover fraction? Line 100: You should drop "imagery" if you are talking about instruments. You should also add one space before "(EO-1)" in the previous line. Line 104: I would rewrite this sentence as: "to partly take into account irregular shape impacts on snow reflectance". Lines 106 to 108: I am not sure if this information is relevant in this paragraph. Lines 108 to 109: It is unclear what the classification system is for. Is it for classifying SPS? If yes, it would be a good idea to specify that and explain how the system classifies SPS. Lines 130 to 131: In this sentence, the last "retrieval" is redundant. For the sake of conciseness, try to avoid doing this in other sentences of the paper. Line 150: Try to be more specific when mentioning pieces of the part 1 paper. You could maybe mention the section number to help the reader find it in the other paper. Line 152: I would suggest changing the title of the point to: "Difference between field-measured and satellite-derived SPS". Line 183: The acronym BRDF was not introduced in the text yet. Line 189: Try to be consistent with the terminology. I have seen in-situ, field-based, field-measured, and ground-based, used interchangeably. Better if you choose the most appropriate and use it consistently throughout the text. Line 199: European Space Agency (ESA) was previously introduced. Figure 1: It would be good to add a picture of the Senator Beck Basin site as well. In the map, make sure to increase the font and include the name

of the two sites. Ideally, it would be good to have an inset zoomed to the two sites together with the US map. Table 2: I am not sure if the SnowEx17 column is necessary here, since there is no linkage to the Yang columns, and it was previously mentioned. Line 239: The sentence "masking by gases and molecules" is not the most accurate. I would suggest changing it for "attenuation and scattering by gases and aerosols". Figure 2: I would suggest trying to minimize the amount of text in this flowchart. Also, you would have to connect the two biggest green boxes inside the dashed line to represent better that this is an iteration process. Figure 3: I am impressed with the spatial detail of XBAER retrievals, but I noticed two geometrical-shaped features in Eastern Greenland. Could you please comment on why this is happening in that region? Line 336: This parenthesis is probably unnecessary: "(humidity, temperature, . . . etc.)". Lines 336 to 347: I appreciate that you tried to compile as many studies as possible to perform a qualitative validation, but this section is too long. Instead of listing all the values, you can try to summarize what you found by the other authors focusing only on what you used for your validation. Line 348: There is no need to use quotations here. Figure 5: It seems that the legend of the cloud maps is wrong. It should be cloud (light blue) and cloud-free (white). Line 529: I am not sure if time series would be the best term to describe this analysis. The Sentinel-3 image is a snapshot of time, while the aircraft surveying takes about 2 hours to complete. It would be better to relate this to space (coordinates), and probably some correction would be needed to address differences in solar zenith angle between the two instruments. Was that addressed? Differences in solar zenith angle can also represent different amounts of shadows, which might explain some of the differences between XBAER and SMART. Lines 533 to 534: The mean SGS from SMART is actually higher than for XBAER. Lines 578 to 579: That depends on environmental conditions. Lines 587 to 589: It is unlikely that blowing snow would transport fresh snow from the ground to such long distances in such a short period. Line 636: There is extra space before the parenthesis. In addition, there is no need to repeat the ice crystal types in the conclusions. Line 656: It would be better to use "inversely correlated" than "anti-correlated".

Technical Corrections

Line 54: Replace "of change snow properties" to "of snow properties change". Line 54: Replace "annular" for "annual". Line 60: Replace "temperatures surrounding" for "surrounding temperatures". Line 70: Replace "summery" for "summarize". Line 86: Replace "Jin et al (2008)" for "Jin et al. (2008)". Line 90: Replace "usage" for "use". Line 96: Replace "the in-situ measurement" for "in-situ measurements". This sentence would be clearer if you add a comma after Antarctica. Line 114: Replace "e.g." for "e.g.,". This repeats a couple more times throughout the text. Line 116: There is a missing space before the citation parenthesis. Line 182: Replace "SLSTR/AATSR" for "SLSTR and AATSR". Line 217: There is a missing period. Also, you should replace "have not linkage" for "have no linkage". Line 226: Replace "is" for "was". Line 233: Replace "present" for "presented", and add a comma after "(about 80° SZA)". Line 263: Replace "details" for "detailed". Line 373: Replace "is" for "are". Line 439: Should replace "the warmer conditions leads to" for "which leads to". Line 527: Remove hyphen after "SGS". Line 645: Replace "minimization" for "minimizations". Line 671: Replace "usage" for "use".

---

## Author Comment (AC1) · 5 Feb 2021

Dear Editor, dear Dr. Ghislain Picard,

Thanks for the valuable comments, which help to improve the quality of the paper. The detailed replies are addressed below point by point in blue. In short:

(1)  More validation is included

|  | **Current version** | **Revised version** |
| --- | --- | --- |
| Number of site(s) | 1 | 7 |
| Total observation length | 1 month | ~10 years |

(2) The discussion with respect to snow particle shape, especially under different classification systems in different scientific communities.

To retrieve snow properties from satellite observations, what we need is the local optical properties for an "effective particle shape" to perform the radiative transfer calculation, we will emphasize that we should not over-interpret the effective particle shape we retrieved in the revised version. As highlighted in Picard et al (2009), "information is urgently needed to know which model shape best approximates the different type of fresh snow", to address "the uncertainty of SSA retrieval based on the SSA-albedo relationships when grain shape is unknown". We believe our work, as a first step/attempt, provides some new/useful way/information for this issue. And of course, we will introduce a more sophisticated way in our future work, for example, to mix different shapes.

Picard, G., Arnaud, L., Domine, F. and Fily, M., Determining snow specific surface area from near-infrared reflectance measurements: numerical study of the influence of grain shape, Cold regions and science and technology, 56, 10-17,2009

Best regards,

Linlu Mei on behalf of all co-authors

Review "The retrieval of snow properties from SLSTR/Sentinel-3 -part 2: results and validation" by Mei and colleagues. The paper aims at validating an algorithm to retrieve snow grain size and shape, and snow specific surface area from the space-borne SLSTR sensor. The algorithm was described in another paper in review (companion part 1), the present manuscript is dedicated to the validation. The overall goal of these two parts is of interest for the cryosphere community, in particular because SLSTR is on the Sentinel 3 series of satellite which will be able for decades.

The paper is original and clear. Nevertheless, my recommendation is to postpone the acceptance of this paper for three main reasons:

Response: Thanks for the very valuable comments from Dr. Ghislain Picard, after detailed discussion with him by emails, we hope we have a good understanding of all comments here. The key issue, as raised by Dr. Ghislain Picard, is about more validation. This is also raised by the second reviewer of part 2. Although, as mentioned by the reviewer, "I understand that it is hard to obtain enough data for remote sensing validation", we have started to collect more validation data since we saw the comments on 10 Nov. 2020.

The reason why only SnowEx17 was considered for the validation in the current form, is that, to our best knowledge, this is possible the only campaign providing all three satellite retrieved parameters (SGS, SPS and SSA). Now, the new understanding is that we can use any campaign data, even when only one satellite retrieved parameter is provided in the campaign. Thus, the following campaign data have been collected for an enhanced validation, in short, the validation is largely extended from one single month from the SnowEx17 campaign to a couple of years worldwide (see Fig 1). We believe, that the extended validation will provide a comprehensive understating of the performance of XBAER algorithm.

[Figure]

Fig. 1 Geographic distribution of the validation sites. The colors represent the type of each site while the observation period used in this manuscript is indicated near each site.

Please be noted that the above campaign data, covering all typical snow-covered geographic regions, will also provide deeper understanding of potential atmosphere/surface effects. For instance, if we make a cross-validation between the Japanese site and Dome C site, we may have a much better understanding of the impact of aerosol contamination, while the comparison between French Alps and North

America may provide more information of the impact of surface elevation.

The validation is based on a too limited set of in-situ data, part of it is discarded because of cloud contamination (SnowEX17). The text truly dedicated to the algorithm performance evaluation is also relatively short and seems unfinished, most of text is about the difficulty to perform the validation, which in the end does not contribute to give confidence in the retrieval algorithm. The conclusion about the algorithm performance therefore lacks of support. There are also several technical issues (see below in the detail comments) in particular one on the RMSE definition. The lack of datasets is a common problem, but not to the extent depicted by the authors. The main example is for the snow grain size. The manuscript cites Kokhanovsky et al. 2019 which pursues a very similar objective as the present manuscript but uses OLCI, (on Sentinel 3 as well as SLSTR) to estimate grain size and SSA (not the grain shape). For the validation these authors used an extensive dataset with 100s of SSA field measurements in Greenland and in Antarctica. These data can be either retrieved from the graphs or in principle obtained from the authors, and should be used here to complete the validation (or even replace the 3 SnowEx measurements). Moreover, the performance between the SLSTR and OLCI algorithms could be analyzed at these in-situ points. At last, the authors "emphasize that the results presented in this section is considered as preliminary" (L373). They indeed propose to include Mosaic data in their analysis in the future. My concern is whether it is worthwhile for the community to publish "preliminary results" in two papers. My suggestion is indeed to wait for complete results and include Mosaic dataset.

Response: We have contacted the MOSAiC team and we will have to wait for quite long for the processing of the data, however, as we mentioned above, thanks for all snow scientists who are willing to share the valuable dataset, we have collected enough campaign data for an extended validation.

We also add the comparison over Greenland with the retrieval from OLCI (Kokhanvosky et al., 2019) in the revised version. However, SSA retrieved in Kokhanvosky et al. (2019) used the simple relationship between SGS and SSA, that is

$SSA = \frac{3}{\rho \times SGS}$ , ρ is the bulk ice density. Even the SGS is perfectly retrieved, the

calculation using this simple "conversion" may provide 20% error, and the SSA-albedo relationships limits the accuracy of SSA retrievals from albedo when the grain shape is unknown (Picard et al., 2009). We believe that our work is a new attempt to provide the information we are lacking now, that is we retrieved an "effective particle shape" and SGS, and provide unique relationship between SSA and SGS.

For instance, in the case of convex faceted particles such as droxtal, solid column, and plate, the calculation of total area is straightforward and based on the Cauchy's surface

area formula:

$$A = 4A_p.$$

(1)

Taking into account that for selected SPS, one can find corresponding V and $A_p$ in database given by Yang et al., (2013), we have the following results for SSA of such particles:

$$SSA = \frac{4A_p}{\rho V}.$$

(2)

In this case a solid column includes two equal cavities in the form of a hexagonal pyramid and cannot be considered as convex particle. The aspect ratio of hollow column with the height, d, of hexagonal pyramid is given according to Yang et al., (2013) as:

$$\frac{2a}{L} = \begin{cases} 0.7, & L < 100\,\mu m \\ \dfrac{6.96}{\sqrt{L}}, & L \ge 100\,\mu m \end{cases}, \quad d = 0.25L.$$

(3)

The volume of such hollow column is given by

$$V = V_c - 2V_p,$$

(4)

where the volume of solid column, $V_c$, and a hexagonal pyramid, $V_p$, are,

$$V_c = \frac{3\sqrt{3}}{2} a^2 L,$$

(5)

$$V_p = \frac{\sqrt{3}}{2} a^2 d.$$

(6)

Thus, the volume, V, is

$$V = \frac{\sqrt{3}}{2} a^2 (3L - 2d).$$

(7)

Employing the relationship between d and L given by Eq (A4) and excluding a, we have

$$V = \frac{2.5\sqrt{3}}{2} a^2 L \begin{cases} m_0 m_1^2 L^3, & L < 100\,\mu m \\ m_0 m_2^2 L^2, & L \ge 100\,\mu m \end{cases},$$

(8)

where $m = \dfrac{2.5}{\sqrt{3}/2}, m_1 = \dfrac{0.7}{2}$, and $m = \dfrac{6.96}{2}$. For a selected volume, V, the length, L, is calculated as follows:

$$L = \begin{cases} [V / m_0 / m_1^2]^{\frac{1}{3}}, & V < V_{100} \\ [V / m_0 / m_2^2]^{\frac{1}{2}}, & V \geq V_{100} \end{cases},$$

(9)

where $V_{100} = m_0 m_2^2 100^2$.

Let us now calculate the area of each triangle side of the pyramid

$$S_t = \frac{a}{2}\sqrt{d^2 + \frac{3a^2}{4}}.$$

(10)

The area of lateral surface of two pyramids is

$$S_p = 3a\sqrt{4d^2 + 3a^2}.$$

(11)

And the total surface area of hollow column is given by

$$S = 6aL + 3a\sqrt{4d^2 + 3a^2},$$

(12)

where a and d should be expressed via L according to Eq. (3).
Having obtained the total area, one can calculate specific surface area

$$SSA = \frac{S}{\rho V},$$

(13)

For each pre-defined effective shape, such a solid derivation is provided in part 1.

Then the key issue becomes can we use the Yang shapes (effective particle shape) to re-produce the real snow properties, which is also raised in the next comment, and our answer is yes and please see detailed explanations and corresponding figure in the next comment.

The issue with respect to the definition of RMSE, is clearly explained in the specific comments later as well.

Picard, G., Arnaud, L., Domine, F. and Fily, M., Determining snow specific surface area from near-infrared reflectance measurements: numerical study of the influence of grain shape, Cold regions and science and technology, 56, 10-17,2009

The grain shape is a big issue of this study. It is claimed to be a major advantage compared to other algorithms (e.g. L617) but the demonstration is missing. First because it is difficult if not impossible to validate. I acknowledge that snow shape is a difficult topic. However as for the validation of the grain size, the choices of the authors are limiting the ability to perform the validation. The algorithm assumes and retrieves geometrical shapes that are representative of precipitating crystals, not of snow on the ground although the algorithm is supposed to be used for snow on the

ground. A first consequence is that the algorithm can not perform well, because the phase function of such shapes does not apply to snow on the ground (expect for fresh snow). Snow on the ground is usually more rounded and irregular than crystals in the atmosphere. The second consequence (and the main one) is the difficulty to perform the validation. Data recorded by snow practitioners and scientists in the field usually follows the international classification of seasonal snow on the ground (Fierz et al. 2009, not cited in the manuscript) which has some shortcomings but is widely used. Since the algorithm does not use these "standard" shapes, it is inherently impossible to perform a fair comparison with external data. It follows a third consequence about the usefulness of the shape information retrieved by the algorithm. I'm wondering how useful is this retrieved "grain shape" for snow community since it does match with its standards. I suggest that to solve this major issue, ideally by adapting the shapes used by the algorithm, and if not possible at least by establishing a link between the different shape systems. Even if imperfect and highly uncertain, this link will benefit to the whole clarity of the paper and will help to shorten the validation section (see comments below). They should also explain why retrieving the shape is useful for the algorithm. The algorithm uses a first guess grain size from another algorithm but no comparison is given. I would expect the authors to demonstrate that taking into account the grain shape has an effective positive impact on the SSA or grain size estimates. This would be very useful for the snow remote sensing community to know if such an approach is fruitful.

Response: We agree that it is not possible for an apple-to-apple validation for the snow grain shape, as we discussed with Dr. Ghislain Picard by emails. Dr. Ghislain Picard also mentioned the way without an assumption of grain shape, that is to use an assumption of stochastic medium, consisting of irregular ice grains and air bubbles, however, in this manner, there is also parameters which cannot be validated. In particular, this is the mean photon path length. It is worth to notice that, all manners, for the retrieval of snow properties from satellite, needs to make some assumption, which is fundamentally needed for a specific retrieval algorithm (Langlois et al., 2020). We extend our introduction part to make a clearer statement in the revised version.

For the widely used ART model (the one used in the retrieval of OLCI in Kokhanovsky et al., 2019), even though the users do not highlight the issues linked to snow particle shape, these issues exist. (1) The original ART model (Zege et al., 2004; Kokhanovsky and Zege et al., 2005) is derived based on the assumption of second-generation fractal for ice crystal shape. (2) In the updated ART model (Kokhnaovsky et al 2018), $g$ and $B$ parameters are introduced. The $g$ parameter depends on both size and shape. The $B$ parameter depends strongly on the shape (Libois et al., 2014). Even one can state that the $g$ and $B$ parameters can be fitted to real observations, several issues linked to the assumption of particle shape occur (1) the accuracy of use single $g$ parameter to describe the complicated particle phase function needs to be checked; (2) ART model

is designed for medium with weakly absorption properties, thus it cannot be used for certain particle size/shape, especially for long wavelength, e.g. 1.6 μm. So, we cannot really avoid making certain (explicit or hidden) assumptions of SPS if it is not iteratively retrieved in the algorithm, like in our case.

To "demonstrate that taking into account the grain shape has an effective positive impact on the SSA or grain size estimates", the mathematical derivation (see example above) is included in part 1 and corresponding sensitivity study is also performed, in the revised version.

The question with respect to if the recent development from Yang can be used for the description of snow properties, such as the snow phase function, this has been confirmed by recent publications (e.g Saito et al., 2019; Pohl et al., 2020; Mei et al., 2021) and private communication with Prof. Ping Yang's group. We have included a detailed explanation in Part 1 and we will make a short summery of this issue in Part 2 as well. Additionally, we have compared the model from Yang with real surface BRDF measurements, including ground-based measurements, aircraft measurements and satellite observations, all shows that Yang shapes can provide good accuracy to simulate snow directional reflectance (Mei et al., 2021), which is the fundamental basis of our retrieval algorithm. Fig. 2 shows an example of how Yang database can re-produce the NASA Cloud Absorption Radiometer (CAR) instrument observed snow BRDF at the flight height of 200 meter, we will include some of our latest investigation in the revised version as well.

[Figure]

Fig 2 Comparison of NASA CAR instrument observed snow BRDF (upper) and Yang shape simulated snow BRDF (lower) for different wavelengths.

In short, the Yang et al. database can be used to describe the ice crystal local optical properties of snow.

With respect to the classification referring to Fierz et al. (2009), as clearly stated in the document, "we expanded and clarified where necessary but did not include those

most recent developments that are not fully agreed upon by the whole community." And as far as I understand, the classification is a work to provide "the creation and maintenance of a common language", no local optical properties are available for the proposed names/classifications. And what we need is the local optical properties for an "effective particle shape" for the RTM calculations. We will emphasize that we should not over-interpret the shape we retrieved in the revised version.

However, we will include certain suggestion of the linkage between Yang's shape and the shapes proposed in Fierz et al. (2009), as suggested by Dr. Ghislain Picard. We are currently harmonizing this issue with Yang's group.

| Fierz et al. (2009) | | Yang et al (2013) |
|---|---|---|
| Precipitation Particles | | Aggregate of 8 columns |
| Machine Made snow | | Droxtal |
| Decomposing and Fragmented | ? | Hollow bullet rosettes |
| precipitation particles Rounded | | Hollow column |
| Grains | | Plate |
| Faceted Crystals | | Aggregate of 5 plates |
| Depth Hoar | | Aggregate of 10 plates |
| Surface Hoar | | Solid bullet rosettes |
| Melt Forms | | Column |
| Ice Formations | | |

We believe that the only way to check the accuracy of a retrieval algorithm is comparison with independent ground-based measurements for parameters such as SGS and SSA, so in our revised version, with such a large validation samples, we will have a comprehensive understanding of the accuracy of XBAER algorithm.

Fierz, C., Armstrong, R.L., Durand, Y., Etchevers, P., Greene, E., McClung, D.M., Nishimura, K., Satyawali, P.K. and Sokratov, S.A. 2009.  The International Classification for Seasonal Snow on the Ground.   IHP-VII Technical Documents in Hydrology N°83, IACS Contribution N°1, UNESCO-IHP, Paris.

Langlois, A., Royer, A., Montpetit, B., Roy, A., and Durocher, M.: Presenting Snow Grain Size and Shape Distributions in Northern Canada Using a New Photographic Device Allowing 2D and 3D Representation of Snow Grains. Frontiers in Earth Science, 7. doi:10.3389/feart.2019.00347,2020

Mei et al., A new snow bidirectional reflectance distribution function (BRDF) model

in the spectral region between UV and SWIR, in preparation, 2021

the benefits to split the study in two parts is not clear. The paper (part 2) presents the validation of an algorithm that is not described, which raise several questions and make it be difficult to read without reading the other paper (part 1). For the review, I didn't read the part 1 (I just browsed it) to be in the same position as a normal reader. I found that reading part 2 was difficult with many open questions about the algorithm and was sometimes annoying because of a few elusive statements referring to the part 1 without providing information. E.g. "The similarities and differences of the required snow parameters and their accuracy between the snow remote sensing community and other communities (e.g. field-measurement community) are detailed discussed in part 1 of the companion paper (Mei et al., 2020), thus we will not summery again in this paper. ". The length of this part 2 is normal and the information density is relatively low. For the comfort of the reader, I suggest to shorten or remove some sections (e.g. the first results section on Greenland), and merge with the part 1. Only if extending the validation as proposed above with a complete dataset and with Mosaic data, it would be justified to make two papers.

Response: We believe that with comments from reviewers of both part 1 and pat 2, for the revised versions, it is better to keep the two parts separated. The reasons are below:

Besides changes/updates on the current content, the reviewers of Part 1 suggest two more valuable sensitive study, which will further extend the length of the paper. In particular, the new sensitivity study includes

**(1) Impact of spectral response of the two channels at 0.55 µm and 1.6 µm**

In the revised version, one more section to investigate the impact of spectral response of the two channels at 0.55 µm and 1.6 µm is included. The following figure shows the spectral response functions for 0.55 µm (left) and 1.6 µm (right). Using these spectral response functions, we will perform the forward simulation with SCIATRAN model, to get TOA reflectance at 0.55 and 1.6 µm. After that, the retrieval using the XBAER algorithm will be performed. Since in the XBAER algorithm, we did not take the spectral response functions into account, thus this investigation shows the impact of the spectral response function on the retrieval results.

[Figure]

Fig. 3 Spectral response function of 0.55 (left) and 1.6 (right) μm of the SLSTR instrument

**(2) The impact of snow profiles and mixture of different snow shapes**

In order to assess the impacts of snowpack vertical inhomogeneity and the habit mixture on the accuracy of the retrieval algorithm, we add a new section in the revised version. The forward simulation of TOA reflectance at 0.55 and 1.6 µm will be performed using the vertical profile of grain size, particle size distribution, and habit mixture as presented in the following figure. The snow grain size profile was obtained during the SnowEx17 campaign (panel (a)). The particle size distribution of the ice crystal and the habit mixture are provided by Satio et al (2019) (see panel (b) and (c)). Then the retrieval will be performed assuming that the snowpack is vertically homogeneous and consisting of mono-disperse snow particles of single shape, and the retrieval accuracy will be assessed.

[Figure]

Fig. 4 Snow properties used for simulations to investigate the impacts of snow profiles and mixture of different snow shapes on XBAER retrieval (a) snow grain size profile observed during SnowEx17 (b) particle size distribution of snow grain size (c) ratio of snow particle shape. (b) and (c) are suggested by Saito et al (2019)

Saito, M., P. Yang, N. G. Loeb, and S. Kato: A novel parameterization of snow albedo based on a two-layer snow model with a mixture of grain habits, J. Atmos. Sci., 76, 1419–1436, 2019.

And at the meantime, as we mentioned, the validation is also largely extended using almost all available campaign data during 2016 -2020. We believe this extension will satisfy Dr. Ghislain Picard.

We think we always need to make a balance between the overlap content of such companion papers. We have also made a search on snow-topic-related journals, companion papers occur not so often in ground-based community, but very often in the satellite community. For a new retrieval algorithm, a comprehensively theoretical sensitivity study is essentially needed before the retrieval and evaluation of the retrieval results. We will, of course, harmonize again of the overlap content between these two parts. We will make a short summery of the content from part 1, if needed in part2, rather than use "see part 1".

So, in short, we update both parts, by adding new investigations/validations. And we believe that keep them separated is an optimal way, taking both the content and the length into account.

Detailed comments:

L63. What is the definition of "grain size" used here ?

Response: grain size (effective radius) is defined as $3V/(4A_p)$, where $V$ and $A_p$ are the volume and average projected area, respectively.

L 69: correct "detailed discussed"
Response: Done

L70: "summery" → "summary"
Response: Done

L91-L92: I'm not sure to understand "to be with good quality"
Response: "to be with good quality" refers to "the retrieved plane albedo was compared with the measured spectral albedo and a good agreement was obtained with ±10%", stated in the cited paper. We update some details in the revised version.

L98-L99. Please add a reference / name for the operational product.
Response: The product is named as SGSP, which, together with the reference, is included in the revised paper.

L104 I'm not sure to understand "to partly taking snow irregular ".
Response: We removed this sentence in the revised version.

L118: "Details of these issues have been discussed in Part 1 of the companion paper.". Please remove and add a proper reference. Or just remove.
Response: We made a short summary of relevant content from part 1 to part 2.

L120-122: This sentence is strange, "no publication. . . especially using" seems contradictory.
Response: We have updated this sentence in the revision.

L 124-126. I don't understand the sentence. What is an "optimal complex shape". The part 1 paper seems to use very geometrical/simple shapes and the goal of the retrieval

algorithm is to retrieve SPS. How does this apply to this sentence ? Also, what do you mean by the e.g. TOA ?

Response: "optimal complex shape" is the shape for which the difference between simulated and measured reflectance is minimal. That means, we need to pick up 1 "optimal complex shape" from the 9 "candidate shapes".

TOA, as we mentioned in the manuscript, is the Top Of the Atmosphere, the TOA reflectance or radiance is the quantity observed by satellite, which is used later for our retrieval.

We believe the word "complex" is misleading and we deleted this word in the revised version.

L147-149. I suggest to move this statements to the conclusion.

Response: Done

L150. I suggest to remove this statement or merge the two papers.

Response: We included a short summary in part 2 of "the three points we mentioned in Part 1".

L152 – L162. I suggest to move this paragraph to the discussion because it is a typical analysis of the uncertainties of the results/validation. The representativeness issue is a general problem, that affects any in-situ vs remote sensing comparison. Why the SPS would be particular? This also concerns SGS and SSA.

Response: According to our previous experience with non-experts or even experts for the discussion of the comparison between ground-based measurement and satellite retrievals, it is worth to put some general description as we are doing now, in the very beginning of the paper. The "scale issue" can be more than a "general" problem because this fully depends on your retrieval parameters, especially on the "inhomogeneity" of your retrieval parameters.

We had a long discussion with Dr. Joshua King, and we include an investigation of this issue using the observations over tundra basin. The measurements over tundra basin provides the possibility for such an investigation.

[Figure]

**Campaign Totals**
**Snow pits**: 80
**Snow depths**: 21946
**SnowMicroPen**: 1444

| | Static | Roving | Total | $\overline{AT}$ [°C] | Crew [#] |
|---|---|---|---|---|---|
| November | 6 | 16 | 22 | -20.0 | 3 |
| January | 6 | 19 | 26 | -28.4 | 4 |
| March | 6 | 26 | 32 | -8.3 | 5 |

Fig. 4 Information of measurements over tundra basin (provided by Dr. Joshua King)

L170. Remove double "show".
Response: Done

L215. I suggest to remove SnowEx17 grain shape from the table because it is misleading even with the warning in the legend caption. Instead it is possible to list these grain type in the caption and/or in the main text. Note that the grain type measured by SnowEx17 are not specific to this campaign but refer to the international classification (Fierz et al. 2009).
Response: We think the information of SnowEx17 in this table help the readers for a better understanding of the analysis later. We would like to keep it in Table 1. And we update this table by adding possible suggestions of the linkage between Yang shape and the shapes proposed in Fierz et al. (2009). We are currently discussing it with Prof. Ping Yang's group.

L253 have→are
Response: Done

L276. Could you give a definition of spherical albedo and Lambertian surface albedo ?
Response: The Lambertian surface albedo is defined as the ratio of reflected to incident flux.
The spherical albedo is the fraction of the incident solar radiation diffusely reflected over all directions (albedo of an entire planet).
We include some explanation in the revised version.

L281. Could you indicate the resolution of MERRA ?

Response: MERRA resolution is 1°×1°

L282. Our → a
Response: Done

L339-355. The comparison is very qualitative and referring to generic and broad "classification" of "polar snow" does not bring significant information for this validation, especially because not all the existing references about snow grain shape and size have been taken into account. It must be taken into account that July is warm with a large proportion of the ice-sheet subject to melt, which unequivocally leads to rounded coarse grains very quickly.
Response: We largely extend the validation, as we mentioned above. Some measurements will include more information of the shape information, for instance, the aspect ratio of the ice crystal particle. We also highlight that the reader should not over-interpret the retrieved shape.
The impact of temperature on shape is included in the revised version.

Because the validation can not be done with information that are not available, I suggest to convince the reader that the results are plausible using cross-analyzed external data: use MERRA to separate where the snow is fresh and for which the present discussion in these lines apply fairly well.
Response: We include the cross-validation in the extended validation. And a post-processing to remove "ice and dirty snow" is also be introduced in the additional runs.

Where snow is fresh use successive image to show that SGS increases (and SSA decreases) as predicted by metamorphism (as you suggest, July is interesting for the most rapid metamorphism). use passive microwave (or MERRA or SLSTR thermal channels) to separate where melt is active and where the grains are very likely to be rounded. - use the images next 28 July 2017 to demonstrate that the blue shape for instance in NW Greenland are not due to clouds/aerosols. (I've made this comment before reading the discussion, see further comments below).
Response: We include the above suggestion with respect to the explanation into the revised version.

I also suggest to mask out areas in the ablation zone with ice and dirty snow, as the algorithm does not work in these cases. This should be emphasized.
Response: We include post-processing to remove "ice and dirty snow" in our additional runs.

Fig 3. adding a scatterplot with relevant statistics ($R^2$, RMSE, bias, . . .) is common

for a more quantitative validation. In particular, it would be useful to compute the same statistics with the first guess to really show the benefit of the algorithm.

Response: Scattering plot with relevant statistics is used in the extended validation. We also include these parameters from the first guess in our revised version.

L371. The previous section was titled "Results" but was also a comparison (and validation to some extent). Why not a unique Result section that includes both comparison?

Response: Done

L372. I suggest to remove "validate".

Response: Done

L373.  ground-based/aircraft → ground-based and aircraft

Response: Done

L377- 379. I'd remove this introductory sentence that starts by concluding that the algorithm is good although the actual goal of the present sessions is to perform the validation.

Response: Done

L 385. "time and location" or "times and locations".

Response: times and location

L394. Why the rows are not sorted chronologically as in next figure ? What is the order? Has the gray shade in the last row a meaning ?

Response: We have sorted chronologically as in next figure in revised version. We have removed the gray shade in the revised version.

L398. This is the second "Fig 4". Review numbering. +Please add a scale to the maps.

Response: We have harmonized the figure number and put the scale on the maps in the revised version.

L406. How does this perform in the case of thin clouds ?

Response: There will be risk of remaining cloud contamination in the retrieval.

Fig 4 and Fig 5. I don't understand why two figures ?   If I understand well, Fig 4 is a zoom of Fig 5 ? They should be merged in a single composition using the same symbology / graphic style.

Response: We have merged Fig. 4 and 5 in to one figure in the revised version.

L412-413. "is not correctly avoided". This is a bit confusing. The next sentence is clearer to me but seems to be in contradiction with Table 3 indicating "cloud contaminated snow" for this date (which seems accurate based on Fig 4).

Response: We have updated the order and explanations of Table 3. The sample of 9 Feb. (partly cloudy) is not detected by the cloud screening while the sample of 11 Feb. has been detected, however, to check the impact of cloud contamination, we have manually "removed" the cloud screening for sample 11 Feb.

L388 indicates that the comment in Table 3 is obtained with the algorithm. Please clarify.

Response: Done

L413. Give → gives.

Response: Done

L421. Add a ref to the study.

Response: Done

L442. "Our → a" or "our calculations with"

Response: Done

L444. Fig 1 → fig 1 with a lowercase as it is referring to another paper. Add the ref.

Response: Done

L451-452. "cloud effective radius" → "cloud ice crystal effective radius". SGS and "ice crystal size" are used interchangeably in the paper which is sometimes (and especially here) confusing.

Response: We harmonized the names in the revised version.

L464. ""This is similar to the issue in field measurements." what do you mean ?

Response: For the field measurement of SSA, certain shape assumption is also used, and the assumption may not exact occur as well.

Leppanen, L., Kontu, A., Vehvilainen, J., Lemmetyinen, J. and Pullianinen, Comparison of traditional and optical grain-size field measurements with SNOWPACK simulation in a taiga snowpack, Journal of Glaciology, 61, 151-162, 2015

Langlois, A., Royer, A., Montpetit, B., Roy, A., and Durocher, M.: Presenting Snow Grain Size and Shape Distributions in Northern Canada Using a New Photographic Device Allowing 2D and 3D Representation of Snow Grains. Frontiers in Earth Science, 7. doi:10.3389/feart.2019.00347,2020

L465. " (e.g.,the measurement of SSA),". This is generally not true. Do you refer to a precise device and processing?

Response: Yes, this depends on the device and how the measurements are obtained, we include this explanation in the revised version.

L466-470. I'd suggest to define in the method section (Table 2) the most-likely correspondence between Yang's shapes and the snow type defined in the international classification (that used in SnowEx) so it is possible here and in the Section 4 in the results section to assess the algorithm performance in a more rigorous way.

Response: Firstly, we try to make possible linkage between Yang shapes and the "international classification". Secondly, other campaigns (such as campaign performed in China) provide some information with respect to the aspect ratio of particles, which is used to quantify the "accuracy" of shape as well. But again, we would like to highlight that we should not over interpret the retrieved "effective particle shape".

L473. "A previous publication" or cite more than one

Response: Done

L474 are → is

Response: Done

L479 "is 'facet' while XBAER says 'droxtal' both tend to be roundish". Facets according to Fierz et al. 2009 is not rounded. If the retrieval algorithm SPS can not distinguish rounded grains from faceted grains because both are droxtal, how useful it this for field practitioners ? This asks an important question that is not addressed in the introduction: why and for what usage to retrieve SPS from satellite ?

Response: We try to make a linkage between Yang's shape and shapes defined in Fierz et al. (2009). And we will highlight that we should not over-interpret the retrieved SPS in the revised version. The retrieved SPS is an "effective shape", which provides the best agreement between radiative transfer simulations and satellite observations.
As we mentioned above, SGS and SPS are the two fundamental inputs for the RTM calculations in XBAER algorithm.
Additionally, with the extended comparison, we will focus more on the validation of SGS and SSA, in the revised version. The comparison of SPS will be reduced.

L483. I do not agree. It is also believed that grains get rounded due to sublimation in blowing snow (Domine, 2009). This probably depends on the conditions, on the actual

grains available on the surface, and the strength and duration of the saltation/reptation process.

Response: We have updated this statement in the revised version.

L493-496. Please indicate the number of points of each comparison (n=...) and the statistical significance of the results. By "difference" do you mean "rms difference" or "difference of the average" ?

Response: Number of points will be included in the extended validation.

"difference" means "difference of the average"

L548. Here it would be particularly interesting to see how good the first guess predictor of SGS. I'm really interested by knowing if the algorithm sophistication is worthwhile.

Response: We include a small validation/analysis of the accuracy of the first guess.

L533. I'm not sure to understand how the RMSE is calculated. The RMSE includes both systematic and random errors, and here given the difference of the mean, the RMSE should be at least $165 - 138 = 17$ microns while the text indicate 12 microns. Please check also "lower grain sizes".

Response: The definition of RMSE is calculated for two groups (satellite retrievals and corresponding SnowEx measurements), not for one group. The understanding reviewer mentioned above is to calculate RMSE for a single group, which indicates the "scattering properties" of this group of data. In our manuscript, RMSE is calculated as following:

$$RMSE = \sqrt{\frac{1}{N}\sum_{n=1}^{n=N}(SSA_n^{XBAER} - SSA_n^{SMART})^2},$$

where N is the number of samples, $SSA_n^{XBAER}$ and $SSA_n^{SMART}$ are the SSA of sample n obtained from XBAER and SMART retrievals.

L550. The same question applies for SSA, with a difference in the mean of 3 m2/kg, it is not possible that the RMSE is 2 m2/kg.

Response: See above

Fig 8. This figure is interesting but should be used earlier in the validation to infer the errors of estimations. I see the following possible artifacts: - The presence of undetected clouds in the NW Greenland. - The dramatic grain size decrease after 28 July in Eastern Greenland (analysis around L588) is very suspicious and stronger evidences are needed to prove that it would be related to a massive drift event, and not to a retrieval artifact. In particular it would be necessary to demonstrate that the wind sustained over 6m/s for a sufficient long period of time to really bring sufficient

quantities of small grains over the considerable distance. - Why grain shape changes so fast between a Droxtal to a column in central Greenland ? Wind is able to drift fresh snow, but in the absence of recent snowfall, if snow was already Droxtal at the surface, wind can not transform it into more elongated crystals. Faceting of grains at such a pace is suspicious. - The Western side is also affected by the grain size change. The shape change is also marked and different from that observed in the Eastern side. Why this is not discussed?

Response: We have included more explanations for Fig.8, especially with the information of wind from ECMWF. The possible reason of blow of fresh snow due to wind or ice crystal change due to temperature are further analyzed.

L590. The weblink does not point to any data. A figure should be added in the supplementary with direction and wind speed.

Response: We have included the wind information from ECMWF in the revised version.

---

## Author Comment (AC2) · 5 Feb 2021

Dear Editor, dear reviewer,

Thanks for the valuable comments, which help to improve the quality of the paper. The detailed replies are addressed below point by point in blue. The key issue raised by reviewer is we need more validation. The following table shows the update of validation in the revised version.

|  | Current version | Revised version |
| --- | --- | --- |
| Number of site(s) | 1 | 7 |
| Total observation length | 1 month | ~10 years |

Best regards,

Linlu Mei on behalf of all co-authors

**General Comments**

This paper describes the results and validation of the XBAER algorithm that retrieves snow grain size (SGS), specific surface area (SSA), and particle shape (SPS) from Sentinel-3 SLSTR instrument. The paper presents the results and evaluate them using the MODSCAG product, in-situ measurements from SnowEx17, and airborne-based retrievals. The validation for cloud-free and partial cloud cover shows promising results from the XBAER algorithm; however, there are some issues related to the validation process and the paper's writing structure.

Response: The validation is largely extended by including all possible existing campaign during 2016-2020. The analysis, including the writing structure is also re-ranged. The extended dataset for validation is shown in the figure below

[Figure]

Fig. 1 Geographic distribution of the validation sites. The colors represent the type of each site while the observation period used in this manuscript is indicated near each site.

Regarding the validation process, the main negative point is that the authors state that these are preliminary results and that they are waiting for more data from the MOSAiC project to increase the number of observations for validation. However, this paper accompanies another paper on the development of the XBAER algorithm. If these are preliminary results, it would make more sense to fit some of these results in the first paper, and then wait for the MOSAiC data to submit a more comprehensive validation in the second paper. I understand that it is hard to obtain enough data for remote sensing validation, but since there is ongoing data collection, the wisest decision will be to wait until the MOSAiC dataset is fully collected.

Response: Beside the extended validation dataset above, we have also contacted the MOSAiC team, however, the latest information is that we properly have to wait quite some time for the dataset, and we believe that the extended validation is enough for a comprehensive understanding of XBAER algorithm. We will move the content with respect to the MOSAiC comparison in the summery part to indicate our future work.

Regarding the paper's writing structure, in general, the paper lacks conciseness. A few general comments about this topic are the following: - There are long sentences on multiple occasions. - The authors should be more careful about using quantitative adjectives when describing their results or other authors' results. I would recommend using the actual number instead. - I would suggest that the authors make a thorough revision of the use of articles, prepositions, and verb agreements in the paper. - The purpose of this paper should be more clearly stated in the introduction. - There is excessive use of quotes. - The discussion paragraph is too long and speculative. There is

a lot of discussion in the results section already. I would recommend the authors to create a section for results and discussion together instead. - The conclusion is too long and with too much redundancy. It should state the main findings, limitations, and future studies, and if it was able to meet the goals of the study. In addition, the main findings in the conclusion should follow the same order that the results are presented.

Response: We have thoroughly improved the presentation in the revised version. More specifically (1) We have updated the introduction part; (2) We have cut long sentences into short ones; (3) we have fixed possible grammar issues (4) We have merged section 4 and 6 to create a "result and discussion section" (5) We have shorten the conclusion part, with the same order as that the results are presented

More detail is provided in the Specific Comments and Technical Corrections sections below.

Other general comments on the scientific soundness of the paper are the following: How have you dealt with the forest in the Grand Mesa site? - One way to improve the SGS validation would be perhaps to extend the validation using more MODSCAG scenes.

Response: We have performed the simple collocation, according to our understanding, the impact of the forest within an SLSTR pixel is mitigated due to the usage of "effective Lambertian albedo". We include MODSCAG for the SnowEx17 validation in the revised paper.

It is not appropriate to use full cloud cover field measurements to validate a remote sensing retrieval that only works with cloud-free conditions. Using partially covered skies might still be a reasonable assumption, as long as the limitations of the retrievals under these conditions are addressed, but not full cloud cover as on Feb. 11, 2017. Retrieving snow properties for full cloud cover only shows that your model is able to characterize the properties of cloud ice crystals, but that has no implications for snow properties on the ground.

Response: With the largely extended validation, this issue will not be so critical. However, we believe that the validation using the fully cloudy scene is also helpful. We have presented our results in the Sentinel 3 validation meeting held last month (hosted by ESA/EUMETSAT), and most snow scientists show great interest to see how we can avoid a pre-cloud-identification in our snow retrieval because cloud screening above snow always brings large uncertainties in the dataset.

**Specific Comments**

Line 14: The OLCI instrument was not used directly to retrieve snow properties; therefore, it would be better to only mention it when talking about the cloud screening

process.

Response: We removed OLCI here

Line 52: Melting snow does have lower albedo, but the main mechanism that decreases albedo is actually the absence of snow cover.

Response: We updated this sentence in the revised version.

Line 62: The terms field- based and in-situ are synonyms; there is no need to use them together.

Response: We removed field-based in the revised version

Lines 62 to 67: Consider splitting or rearranging this sentence to improve its clarity.

Response: We updated this sentenced and spitted in the revised version.

Line 82: When you mentioned snow fraction. Did you mean snow cover fraction?

Response: Yes, snow cover fraction

Line 100: You should drop "imagery" if you are talking about instruments. You should also add one space before "(EO-1)" in the previous line.

Response: Done

Line 104: I would rewrite this sentence as: "to partly take into account irregular shape impacts on snow reflectance".

Response: Done

Lines 106 to 108: I am not sure if this information is relevant in this paragraph.

Response: We believe it fits here because it gives the reader an overview of the change of SPS with respect to meteorological conditions.

Lines 108 to 109: It is unclear what the classification system is for. Is it for classifying SPS? If yes, it would be a good idea to specify that and explain how the system classifies SPS.

Response: We extended the explanation of the classification of snow from Kikuchi et al (2013) in the revised version.

Lines 130 to 131: In this sentence, the last "retrieval" is redundant. For the sake of conciseness, try to avoid doing this in other sentences of the paper.

Response: We have checked thoroughly of this presentation-related issues and ask our native speaker to double-check as well.

Line 150: Try to be more specific when mentioning pieces of the part 1 paper. You could

maybe mention the section number to help the reader find it in the other paper.

Response: We have made a short summery of information from Part 1, if needed in part 2, rather than use "see part 1" in the current form.

Line 152: I would suggest changing the title of the point to: "Difference between field-measured and satellite-derived SPS".

Response: Done

Line 183: The acronym BRDF was not introduced in the text yet.

Response: It is introduced in the revised version.

Line 189: Try to be consistent with the terminology. I have seen in-situ, field-based, field-measured, and ground-based, used interchangeably. Better if you choose the most appropriate and use it consistently throughout the text.

Response: We harmonized in the revised version.

Line 199: European Space Agency (ESA) was previously introduced.

Response: Removed

Figure 1: It would be good to add a picture of the Senator Beck Basin site as well. In the map, make sure to increase the font and include the name of the two sites. Ideally, it would be good to have an inset zoomed to the two sites together with the US map.

Response: We updated the figures according to the suggestion.

Table 2: I am not sure if the SnowEx17 column is necessary here, since there is no linkage to the Yang columns, and it was previously mentioned.

Response: Together with comments from reviewer 1, we included the classification system from Fierz et al. (2009) in Table 2. The SnowE17 is also based on the Fierz et al. (2009)

Line 239: The sentence "masking by gases and molecules" is not the most accurate. I would suggest changing it for "attenuation and scattering by gases and aerosols".

Response: Done

Figure 2: I would suggest trying to minimize the amount of text in this flowchart. Also, you would have to connect the two biggest green boxes inside the dashed line to represent better that this is an iteration process.

Response: There is an arrow missing which should link the two green boxes.

As to the texts in the flowchart, since XBAER algorithm includes quite some other previous published algorithms, we would like to take the heritage of it.

Figure 3: I am impressed with the spatial detail of XBAER retrievals, but I noticed two geometrical-shaped features in Eastern Greenland. Could you please comment on why this is happening in that region?

Response: Thank you for the positive feedback of our retrievals. The two geometrical-shape features in Eastern Greenland are explained by the impact of large viewing zenith angle. This Figure is created by three SLSTR swaths, the two geometrical-shaped features occur at the edge of the middle swath (large viewing zenith angles). In XBAER, the "effective Lambertian albedo" assumption is used and this assumption will introduce error under large viewing zenith angle condition. We included the explanation in the revised version.

Line 336: This parenthesis is probably unnecessary: "(humidity, temperature, … etc.)".

Response: We removed it

Lines 336 to 347: I appreciate that you tried to compile as many studies as possible to perform a qualitative validation, but this section is too long. Instead of listing all the values, you can try to summarize what you found by the other authors focusing only on what you used for your validation.

Response: We reduced certain previous studies in this section, to make sure that only close-related publications are cited here.

Line 348: There is no need to use quotations here.

Response: We removed it

Figure 5: It seems that the legend of the cloud maps is wrong. It should be cloud (light blue) and cloud-free (white).

Response: The current legend is a little bit mis-leading, the "snow-free" legend refers to the area where XBAER retrieval is not performed, this includes (1) snow-free and cloud free (2) cloud above snow; (3) cloud above snow-free. The light blue is snow. We updated the legend in the revised version.

Line 529: I am not sure if time series would be the best term to describe this analysis. The Sentinel-3 image is a snapshot of time, while the aircraft surveying takes about 2 hours to complete. It would be better to relate this to space (coordinates), and probably some correction would be needed to address differences in solar zenith angle between the two instruments. Was that addressed? Differences in solar zenith angle can also represent different amounts of shadows, which might explain some of the differences between XBAER and SMART.

Response: We have published another paper for this topic (Jäkel et al., 2021), in which more detailed comparison between XBAER and SMART is given. We included some findings from the new publication in the revised version. Specifically, possible shadow

effect on the retrieval accuracy is detailed discussed in the paper of Jäkel et al (2021).

Jäkel, E., Carlsen, T., Ehrlich, A., Wendisch, M., Schäfer, M., Rosenburg, S., Nakoudi, K., Zanatta, M., Birnbaum, G., Helm, V., Herber, A., Istomina, L., Mei, L., and Rohde, A.: Comparison of optical-equivalent snow grain size estimates under Arctic low Sun conditions during PAMARCMiP 2018, The Cryosphere Discuss. [preprint], https://doi.org/10.5194/tc-2021-14, in review, 2021.

Lines 533 to 534: The mean SGS from SMART is actually higher than for XBAER.
Response: We fixed this word in the revised version.

Lines 578 to 579: That depends on environmental conditions.
Response: We included more information for these sentences to clarify the dependent of length of time scale on environmental conditions.

Lines 587 to 589: It is unlikely that blowing snow would transport fresh snow from the ground to such long distances in such a short period.
Response: We included the ECMWF wind information in the revised version, and an updated explanation for this sentence will be included.

Line 636: There is extra space before the parenthesis. In addition, there is no need to repeat the ice crystal types in the conclusions.
Response: extra space before the parenthesis is removed and we have deleted the ice crystal types in the conclusion.

Line 656: It would be better to use "inversely correlated" than "anti-correlated".
Response: Done

**Technical Corrections**
Line 54: Replace "of change snow properties" to "of snow properties change".
Response: Done

Line 54: Replace "annular" for "annual".
Response: Done

Line 60: Replace "temperatures surrounding" for "surrounding temperatures".
Response: Done

Line 70: Replace "summery" for "summarize".
Response: Done

Line 86: Replace "Jin et al (2008)" for "Jin et al. (2008)".
Response: Done

Line 90: Replace "usage" for "use".
Response: Done

Line 96: Replace "the in-situ measurement" for "in-situ measurements". This sentence would be clearer if you add a comma after Antarctica.
Response: Done

Line 114: Replace "e.g." for "e.g.,". This repeats a couple more times throughout the text.
Response: Done

Line 116: There is a missing space before the citation parenthesis.
Response: We have included a space

Line 182: Replace "SLSTR/AATSR" for "SLSTR and AATSR".
Response: Done

Line 217: There is a missing period. Also, you should replace "have not linkage" for "have no linkage".
Response: The problems are fixed and phase updated.

Line 226: Replace "is" for "was".
Response: Done

Line 233: Replace "present" for "presented", and add a comma after "(about 80° SZA)".
Response: Done

Line 263: Replace "details" for "detailed". Line 373: Replace "is" for "are".
Response: Done

Line 439: Should replace "the warmer conditions leads to" for "which leads to".
Response: Done

Line 527: Remove hyphen after "SGS".
Response: Done

Line 645: Replace "minimization" for "minimizations".

Response: Done

Line 671: Replace "usage" for "use".
Response: Done

---

## Editor Decision (ED1)

2021-02-11

Submission tc-2020-270

Thank you for your submission to be considered for publication in The Cryosphere. The authors did significant amount of work to improve from the initial version of the paper, with a significant amount of new data that was added to enhance the validation. The new data also increases the validation period which was a key element to be improved form the initial submission.

A main point of concern for me is the SPS derived information. On fig 3, it is suggested that SPS of columns are present at the surface on Greenland. I don't see how this would be possible simply from a metamorphic perspective in such environment. Especially since it covers a very large portion so a clear explanation on why columns would be found on the surface is required. Same is observed on fig 7 where columns are derived in northern Canada and Russia which is not plausible. The SPS really needs better description, given also that most of the Canadian Arctic Archipelago is covered with aggregates of 8 columns which is simply not the case.

At this stage, I will provide here several minor comments that may help further polish the manuscript.

- The introduction in this paper includes more motherhood material on the importance of snow that I think should be moved and used into Part 1 companion paper. Part 1 paper introductions falls short in explaining the 'snow problem' from a remote sensing perspective, and the impact climate change has on snow. However, this is described in Part 2, where I think it should be the opposite. The authors should consider moving text into Part 1 introduction. This would also help reducing the introduction which I think is too long at this stage.
- Fig 1c) is not a detailed example of measured roughness the wording should be changed. The photo is a surface based passive microwave radiometer;

- The text on flowchart has varying fonts, I would suggest removing bold in the green boxes;

Regards,

Prof. Dr. Alexandre Langlois

Associate editor, *The Cryosphere*